# Lack of Paxillin phosphorylation promotes single-cell migration in vivo

Qian Xue[1], Sophia R.S. Varady[1], Trinity Q Alaka'i Waddell[1], Mackenzie R. Roman[1], James Carrington[1,2], and Minna Roh-Johnson[1]

**Focal adhesions are structures that physically link the cell to the extracellular matrix for cell migration. Although cell culture studies have provided a wealth of information regarding focal adhesion biology, it is critical to understand how focal adhesions are dynamically regulated in their native environment. We developed a zebrafish system to visualize focal adhesion structures during single-cell migration in vivo. We find that a key site of phosphoregulation (Y118) on Paxillin exhibits reduced phosphorylation in migrating cells in vivo compared to in vitro. Furthermore, expression of a non-phosphorylatable version of Y118-Paxillin increases focal adhesion disassembly and promotes cell migration in vivo, despite inhibiting cell migration in vitro. Using a mouse model, we further find that the upstream kinase, focal adhesion kinase, is downregulated in cells in vivo, and cells expressing non-phosphorylatable Y118-Paxillin exhibit increased activation of the CRKII-DOCK180/RacGEF pathway. Our findings provide significant new insight into the intrinsic regulation of focal adhesions in cells migrating in their native environment.**

## Introduction

Cell migration is fundamentally required during many biological processes, including embryonic development, immune surveillance, and wound healing (Lauffenburger and Horwitz, 1996; Trepat et al., 2012; Horwitz and Webb, 2003; SenGupta et al., 2021). Uncontrolled cell migration leads to diseases such as cancer metastasis and inflammation (Friedl and Wolf, 2003; Luster et al., 2005). Therefore, understanding the mechanistic basis of cell migration is critical both for fundamental aspects of biology as well as the pathology of diseases.

Focal adhesions are macromolecular structures that physically link the cell cytoskeleton to the outside extracellular matrix (ECM) during cell migration (Wozniak et al., 2004; Lauffenburger and Horwitz, 1996). The dynamic assembly and disassembly of focal adhesions at the front edge and trailing edge of the cell, respectively, are tightly coupled with actomyosin contractility to facilitate efficient cell migration (Vicente-Manzanares and Horwitz, 2011). During migration, focal adhesions are necessary for mechanical force sensation and generation, as well as serve as signaling hubs that transduce signals to the cell and the environment (Balaban et al., 2001; Zaidel-Bar et al., 2007a; Turner, 2000; Eke and Cordes, 2015; Hoffman, 2014; Oakes and Gardel, 2014; Doyle et al., 2022). Over the past 40 years, numerous proteins have been discovered and identified as focal adhesion proteins (Zaidel-Bar et al., 2007a; Legerstee and Houtsmuller, 2021; Wozniak et al., 2004; Wu, 2007; Burridge, 2017). Furthermore, 3D super-resolution microscopy techniques reveal that focal adhesions have multilaminar protein architecture containing at least three interdependent spatial and functional protein strata, including the integrin signaling layer, force transduction layer, and actin regulatory layer (Kanchanawong et al., 2010). Thus, focal adhesions are well-coordinated protein structures to regulate cell migration.

Most mechanistic studies of focal adhesions are based on in vitro cell culture assays where focal adhesions can be readily visualized at the ventral surface of cells with live cell microscopy. These mechanistic studies have been enormously valuable and have identified Paxillin as a key component in focal adhesions. Paxillin acts as a scaffold in focal adhesions, transmitting extracellular signals into the intracellular space and activating signaling cascades required for cell migration (López-Colomé et al., 2017; Turner et al., 1990; Glenney and Zokas, 1989). Paxillin activation is tightly regulated by the phosphorylation status of several tyrosine and serine residues along the length of the protein (Burridge et al., 1992; Schaller and Schaefer, 2001; Bellis et al., 1997). Following integrin activation, tyrosine kinases such as focal adhesion kinase (FAK) are activated and, in turn, phosphorylate Paxillin at tyrosine 118 (Y118) and tyrosine 31 (Y31; Mitra et al., 2005; Bellis et al., 1995; Schaller and Parsons, 1995). Phosphorylated Paxillin then recruits the SH2/SH3 adaptor protein, CRKII, leading to downstream GTPase activation and induction of cell motility (Lamorte et al., 2003; Petit et al., 2000; Birge et al., 1993; Brugnera et al., 2002; Kiyokawa

---

[1]Department of Biochemistry, University of Utah, Salt Lake City, UT, USA;   [2]School of Medicine, University of Utah, Salt Lake City, UT, USA.

Correspondence to Minna Roh-Johnson: roh-johnson@biochem.utah.edu.



et al., 1998b; Schaller and Parsons, 1995; Tsubouchi et al., 2002; Vallés et al., 2004; Abassi and Vuori, 2002). Previous research also reveals that phosphorylation of Paxillin at Y118 and Y31 increases the rate of focal adhesion disassembly, thus promoting faster focal adhesion turnover and membrane protrusions in migrating cells (Zaidel-Bar et al., 2007b). Thus, tyrosine phosphorylation of Paxillin is a key event in controlling focal adhesion dynamics and overall cell migration.

Focal adhesion regulation has been less well-defined in 3D environments. Initial experiments in 3D matrices generated a controversy as to whether cells even formed focal adhesions during cell migration in 3D (Fraley et al., 2010). Follow-up experiments revealed that cells do indeed form focal adhesions during cell migration in 3D environments (Deakin and Turner, 2011; Kubow and Horwitz, 2011; Doyle et al., 2015; Geiger et al., 2009; Geraldo et al., 2012; Yamada et al., 2003; Yamada and Sixt, 2019; Harunaga and Yamada, 2011; Doyle and Yamada, 2016). Despite these advances in understanding focal adhesion-based migration in more complex 3D environments, it is still largely unknown how focal adhesions form and function in an in vivo environment, in which the mechanics, signaling, and cellular interactions are intact. Focal adhesions have been described during collective cell migration, in which sheets or groups of cells migrate or change their cell shape as a collective, and these processes have been beautifully dissected with high-resolution imaging approaches in animal models (Yamaguchi et al., 2022; Olson and Nechiporuk, 2021; Gunawan et al., 2019; Goodwin et al., 2017; Goodwin et al., 2016; Fischer et al., 2019; Lewellyn et al., 2013; Bischoff et al., 2021). While these studies illustrate the role of focal adhesions in collectively migrating cells, there is very little known about how single cells use focal adhesion machinery for migration. Single-cell migration is a critical process in cancer metastasis and immune cell recruitment, and despite this importance in disease and pathogenesis, it is still unclear what molecular components make up a focal adhesion in single cells migrating in an in vivo environment, let alone the intrinsic regulation of these molecular players. One live-imaging study in zebrafish showed Paxillin-positive punctate structures in migrating macrophages (Barros-Becker et al., 2017), revealing that single cells potentially form focal adhesions during cell migration. However, the dynamics and intrinsic regulation of focal adhesion biology are still entirely unclear during cell migration in a physiologically relevant, intact environment.

A major limitation to understanding focal adhesion formation and regulation during single-cell migration in vivo is the lack of model systems where transient subcellular focal adhesion structures form efficiently and can be readily visualized under high-resolution imaging in live animals. Here, we developed a zebrafish cancer cell transplantation system in which we can directly visualize focal adhesion structures during single-cell migration in vivo. Similar to other animal models, this in vivo system does not easily allow for the manipulation of environmental components to determine which specific factors in the environment dictate focal adhesion regulation. However, this system does allow for the visualization and analysis of focal adhesion formation and regulation with high resolution in an unperturbed, physiologically relevant in vivo environment. Taking advantage of this in vivo system, we aimed to determine whether we could dissect the molecular regulation of focal adhesions in single cells migrating in their native environment as compared with the traditional in vitro cell culture system. Surprisingly, we found that a key focal adhesion protein, Paxillin, exhibits differential molecular dynamics in cells in vivo than in vitro. Furthermore, we found that Y118-Paxillin exhibited significantly reduced phosphorylation in migrating cells in vivo, despite being a key phospho-site for cell migration in vitro. To directly test the function of Y118-Paxillin, we performed site-directed mutagenesis studies and found that surprisingly, non-phosphorylatable Y118-Paxillin promoted cell migration in vivo, despite inhibiting cell migration in vitro, and non-phosphorylatable Y118-Paxillin increased rates of focal adhesion disassembly in vivo. These results provide a previously undescribed mechanism for the intrinsic regulation of focal adhesion formation in cells migrating in their native environments.

## Results

### An in vivo system to visualize focal adhesion dynamics

To study single-cell migration in vivo, we took advantage of the optically transparent larval zebrafish due to the ease of cellular visualization in an intact organism. Since cell migration is a highly dynamic process that is tightly controlled by many environmental factors, we used a syngeneic (same species) transplantation approach in the zebrafish system, allowing cells to migrate under physiological conditions. We transplanted highly migratory zebrafish melanoma cells, ZMEL cells, into the hindbrain ventricle of the larval zebrafish 2 d postfertilization (dpf). The larval hindbrain takes up transplanted cells readily and contains a variety of components that exist in the tumor microenvironment, such as immune cells, vasculature, ECM, and other supporting cells (Roh-Johnson et al., 2017). Over the course of 1–4 d post-transplantation, ZMEL cells disseminated into different larval tissue, such as the skin, neuroepithelial tissue, muscle, and tail fin (Fig. 1 A). To study focal adhesion biology during single-cell migration in vivo, we aimed to identify an in vivo microenvironment where ZMEL cells use focal adhesion-based migration that is also amenable to live animal imaging. Of the regions in which ZMEL cells disseminated, we focused primarily on the skin (Fig. 1, B and C) as the skin is known to be enriched in ECM, a substrate that is required for focal adhesion formation. Furthermore, the skin is the relevant tissue environment for melanoma cell behavior and it is accessible for high-resolution microscopy due to its superficial location in the animal.

We sought to determine whether ZMEL cells form focal adhesions when migrating in the skin. Thus, we first determined whether ZMEL cells migrating in the skin are in physical contact with the ECM. Using immunofluorescence assays, we found that ZMEL cells embedded in the skin are in close proximity with laminin (Fig. 1 D, upper panel), a protein previously reported to be enriched in skin ECM (Jessen, 2015). Live microscopy with a collagen reporter $Tg(krt19:col1a2\text{-}GFP)^{zj502}$ (Morris et al., 2018)

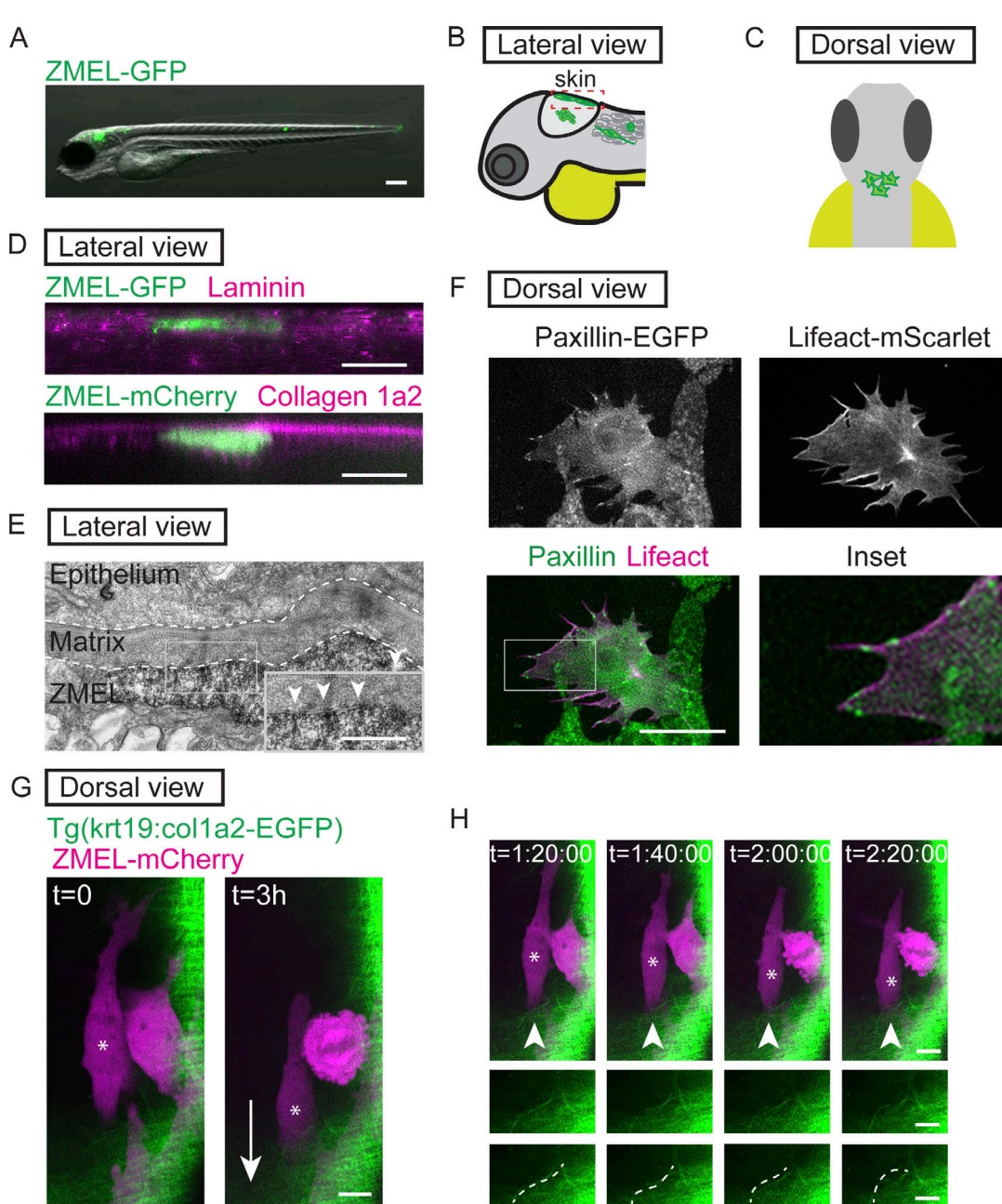

Figure 1. **Transplanted ZMEL cells form focal adhesions structures during single-cell migration in vivo. (A)** Representative image of ZMEL-GFP whole body dissemination in a 5 dpf (3 d post-transplantation) larval zebrafish. Scale bar is 100 μm. **(B and C)** Schematic of two imaging views to visualize transplanted ZMEL cells that attach to the zebrafish skin. Lateral view (B); dashed red box indicates skin region; dorsal view (C). **(D)** Upper panel: Lateral view of a fixed zebrafish larva with transplanted ZMEL-GFP cells (GFP immunostaining, green) in close proximity with laminin (magenta). Lower panel: Lateral view of a live larva with transplanted ZMEL-mCherry cells (pseudocolored in green) that is proximal to collagen labeled with *Tg(krt19:col1a2-GFP)*[zj502] (pseudocolored in magenta). Scale bar is 10 μm. **(E)** TEM micrograph of a ZMEL cell transplanted in a larval zebrafish (3 d post-transplantation), lateral view. Dashed white lines outline the skin ECM, with a pigmented ZMEL cell underneath the matrix (labeled "ZMEL"). The inset is a magnification of the grey box revealing the ZMEL–matrix interface. Arrowheads mark the electron-dense regions where ZMEL cells contact the matrix. Scale bar is 1 μm. See also Fig. S1 B for non-ZMEL containing control larvae. **(F)** Live imaging of transplanted ZMEL cells co-expressing zebrafish Paxillin-EGFP (green in overlay and inset) and Lifeact-mScarlet (magenta in overlay and inset) in the zebrafish skin. Inset is the magnified image of the grey box in the overlay. Dorsal imaging view. Scale bar is 10 μm. See also Video 1. **(G)** Start and end frames from a timelapse video of transplanted ZMEL-mCherry cells (magenta) migrating in the zebrafish skin with collagen (green) labeled with *Tg(krt19:col1a2-GFP)*[zj502]. Arrow indicates the direction of migration, and the migrating ZMEL cells are marked with an asterisk. Scale bar is 10 μm. **(H)** Still images of timelapse video in G. Arrowheads mark a collagen fiber (green) that is buckling as ZMEL cells (magenta, asterisk) migrate. Middle row shows the magnified images of collagen fibers. Bottom row highlights the buckling collagen fiber overlaid with a dashed white line. See also Video 2.

revealed that ZMEL cells are proximal to the collagen-positive layer of the skin (Fig. 1 D, lower panel). We also found that larval zebrafish typically do not contain fibronectin in this tissue, but upon ZMEL transplantation, we detected fibronectin surrounding ZMEL cells with immunohistochemical approaches (Fig. S1 A). These results suggest that ZMEL cells, or neighboring cells, might secrete fibronectin when ZMEL cells are transplanted into larvae. Thus, ZMEL cells interact with a combination of ECM components during migration. Due to the resolution limitation of light microscopy, we further performed transmission electron microscopy (TEM) on larval zebrafish with transplanted ZMEL cells. Using non-ZMEL containing larvae as a control for overall skin tissue architecture (Fig. S1 B), we were able to identify ZMEL cells in the skin of transplanted larvae. TEM micrographs indicated that ZMEL cells are in direct contact with the matrices in the skin (Fig. 1 E). Furthermore, we visualized electron-dense structures at the ZMEL–matrix interface (Fig. 1 E, inset), suggesting that focal adhesion-like structures (Medalia and Geiger, 2010) form at these interfaces. Together, these results suggest that ZMEL cells make direct physical contact with the ECM in the skin tissue of the larval zebrafish.

We next sought to determine whether ZMEL cells localize focal adhesion components to their ventral surfaces during single-cell migration in vivo. To visualize focal adhesions in vivo, we used the zebrafish MiniCoopR system (Ceol et al., 2011) to generate zebrafish melanoma tumors expressing zebrafish Paxillin (Fig. S1 C), a core focal adhesion protein, tagged with EGFP. From these zebrafish tumors, we established a primary ZMEL cell line—ZMEL Paxillin-EGFP—using previously established approaches (Heilmann et al., 2015). When ZMEL cells are plated in the in vitro cell culture conditions, Paxillin-EGFP localizes to finger-like protrusions at the ventral surface of ZMEL cells (Fig. S1 C), similar to what is observed in mammalian cells in culture (Turner et al., 1990). These Paxillin-EGFP-positive structures also assemble when ZMEL cells form protrusions and disassemble in regions where ZMEL cells retract. These results suggest that Paxillin-EGFP localizes to focal adhesion structures in ZMEL cells in culture. To determine whether ZMEL cells form Paxillin-positive structures in vivo, we transplanted ZMEL Paxillin-EGFP cells into 2 dpf larval zebrafish and specifically imaged ZMEL cells in the skin. On 1 d post-transplantation, we found that Paxillin forms punctate structures on the ventral surface of ZMEL cells (the surface that makes contact with the ECM in Fig. 1 D) in the skin (Fig. 1 F). To determine whether Paxillin colocalized with another key component of focal adhesions, actin, we generated ZMEL cells co-expressing Paxillin-EGFP and Lifeact-mScarlet and found that Paxillin-positive structures localize along actin fibers in migrating cells in vivo (Fig. 1 F and Video 1). These results suggest that ZMEL cells form Paxillin-positive focal adhesion structures in vivo.

To determine whether ZMEL cells in the skin transduce force to the environmental ECM, we transplanted ZMEL-mCherry cells into the collagen reporter larval zebrafish. We found that as ZMEL cells migrated, a linear collagen fiber started to bend toward the migrating cell in the opposite direction of cell migration (Fig. 1, G and H; and Video 2). These results suggest that

ZMEL cells are actively pulling on collagen fibers as the cells migrate. Together, these results suggest that transplanted ZMEL cells form bona fide focal adhesions during single-cell migration in the zebrafish skin, and we thus took advantage of this unique system of focal adhesion visualization to dissect the mechanics and intrinsic regulation of molecular components of focal adhesions in vivo.

## Paxillin exhibits distinct dynamics in migrating ZMEL cells in vivo as compared to migrating ZMEL cells in vitro

Taking advantage of this in vivo transplantation system, we directly compared the size and Paxillin dynamics between in vivo and in vitro cell culture conditions. We first determined that Paxillin-positive focal adhesions are significantly smaller in cells in vivo compared to cells in culture (Fig. S1 D). We next quantified focal adhesion dynamics—focal adhesion dynamics are determined by the overall lifetime of a focal adhesion protein from assembly to disassembly, as well as the molecular binding dynamics which measures the molecular exchange of a protein between an adhesion and the cytosol (Stehbens and Wittmann, 2014). We first plated the primary ZMEL Paxillin-EGFP cells on in vitro cell culture dishes as well as transplanted the same cell line into 2-dpf larval zebrafish and measured the molecular binding kinetics by fluorescence recovery after photobleaching (FRAP). After photobleaching individual Paxillin-positive structures and monitoring the fluorescence recovery over time (Fig. 2 A and Videos 3 and 4), we found that Paxillin exhibits a significantly faster molecular turnover rate in cells in vivo ($t_{1/2}$ = 6.5 ± 1.7 s) as compared with ZMEL cells in culture ($t_{1/2}$ = 15.5 ± 4.9 s; Fig. 2 B). We next measured Paxillin lifetime at focal adhesions by timelapse microscopy of individual structures (Fig. 2 C; and Videos 5 and 6). From these measurements, we quantified the overall lifetime, as well as assembly and disassembly rates as previously described (Stehbens and Wittmann, 2014; Fig. 2 D). Surprisingly, we did not observe a significant difference in Paxillin lifetime between in vivo and in vitro cell culture conditions (Fig. 2 E); however, Paxillin exhibited significantly faster assembly rates (Fig. 2 F) and slower disassembly rates (Fig. 2 G) in cells in vivo as compared with the in vitro cell culture model. Altogether, these results indicate that the assembly and disassembly rates of Paxillin at focal adhesions differed between the in vivo and in vitro environments.

## Paxillin phosphorylation status is distinct in migrating cells in vivo versus in culture

Since we observed differential assembly and disassembly rates in Paxillin in cells in vivo versus plated under in vitro cell culture conditions, we then sought to identify the molecular regulation that might explain these differences. Paxillin activity has been shown to be tightly regulated by phosphorylation (Burridge et al., 1992), and specifically in large part through tyrosine phosphorylation by an upstream tyrosine kinase, FAK. FAK phosphorylates two tyrosine residues on Paxillin, Y31 and Y118 (Bellis et al., 1995; Schaller and Parsons, 1995), and phosphorylation at these sites regulates focal adhesion disassembly in cell culture models (Zaidel-Bar et al., 2007b). Thus, we sought to investigate whether the differences in Paxillin dynamics

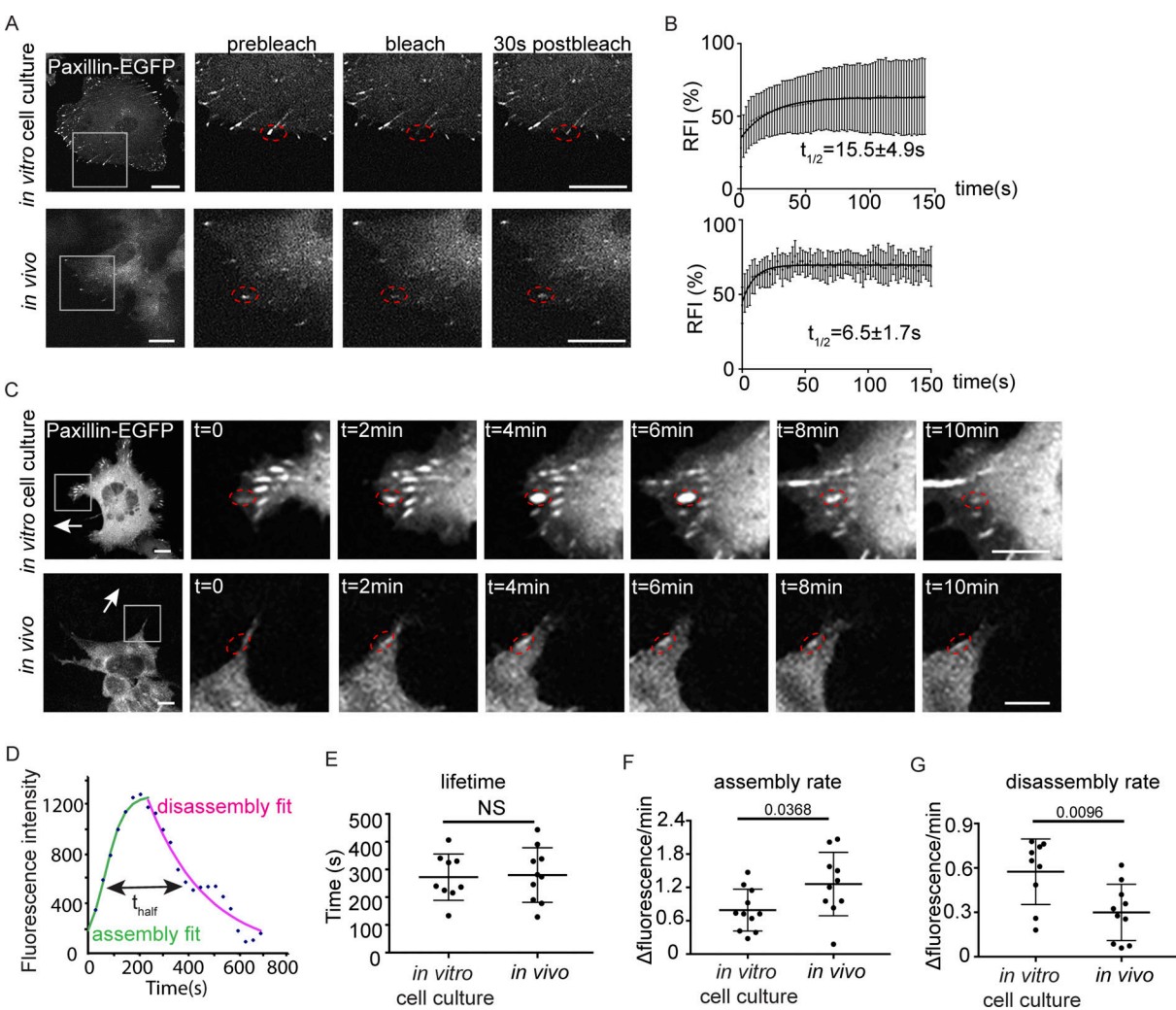

Figure 2. **Paxillin exhibits reduced disassembly rates and increased assembly rates at focal adhesions in migrating ZMELs in vivo. (A)** Still images of ZMEL Paxillin-EGFP FRAP experiments in vitro and in vivo. Left panel is the whole cell view and the rest of the panels are a magnification of the grey boxes prebleach, upon photobleaching, and 30 s after photobleaching. Red dotted circles mark Paxillin positive punctae that underwent photobleaching. See also Videos 3 and 4. **(B)** Cumulative FRAP recovery curves of Paxillin-EGFP in ZMEL cells in the in vitro cell culture conditions and in vivo after photobleaching. $n$ = 36 cells for in vitro, $n$ = 16 cells for in vivo. **(C)** Still images from timelapse videos of ZMEL Paxillin-EGFP, revealing Paxillin lifetimes at focal adhesions in vitro and in vivo. Left panel is the whole cell view and the rest of the panels are a magnification of the grey boxes. Red dotted circles mark the same Paxillin-positive punctae from assembly to disassembly. See also Videos 5 and 6. **(D)** Representative graph of Paxillin lifetime curve fitting in which assembly rate, disassembly rate, and lifetime ($t_{1/2}$) can be calculated. **(E–G)** Quantification of Paxillin lifetime as $t_{1/2}$ (E), assembly rate (F), and disassembly rate (G) in the in vitro cell culture conditions and in vivo from ZMEL Paxillin-EGFP timelapse videos. $n$ = 11 cells for both in vitro and in vivo. Error bars are mean ± SD. Non-parametric unpaired $t$ test. Scale bar is 10 µm.

in vivo are due to the phosphorylation status at these key tyrosine residues. We first tested the phosphorylation status of Y118-Paxillin based on its sequence conservation between zebrafish and other vertebrates (Fig. 3 A), as well as its previously characterized role in cell migration in other zebrafish tissue during morphogenesis (Gunawan et al., 2019; Olson and Nechiporuk, 2021). We examined the phosphorylation of Y118-Paxillin by using a phosphospecific pY118-Paxillin antibody to immunostain ZMEL cells plated on in vitro cell culture dishes, as well as ZMEL cells transplanted in larval zebrafish. As expected, ZMEL cells plated in the in vitro cell culture conditions revealed pY118-Paxillin staining at the ventral surface of the migrating cells (Fig. 3 B, upper panel, white arrowheads; Fig. S2 A, top panel). To determine the Y118-Paxillin phosphorylation status in

migrating cells in vivo, we performed live microscopy to first visualize migrating ZMEL cells in vivo and then processed the same larvae for immunostaining to detect the pY118-Paxillin status in the same migratory cell based on the cell morphology and tissue landmarks. Surprisingly, migrating ZMEL cells in vivo did not exhibit detectable pY118-Paxillin immunostaining (Fig. 3 B, lower panel, white arrowheads; Fig. S2 A, middle panel). We tested for pY118-Paxillin antibody specificity by mutating the Y118 residue and showing a lack of detection by Western blot analysis (Fig. S2 B). As a positive control for pY118-Paxillin immunostaining in vivo, we imaged the zebrafish developing heart (Fig. S2 A, bottom panel), as previous data have indicated positive pY118-Paxillin staining in the heart (Gunawan et al., 2019). Furthermore, we visualized non-ZMEL cells in the

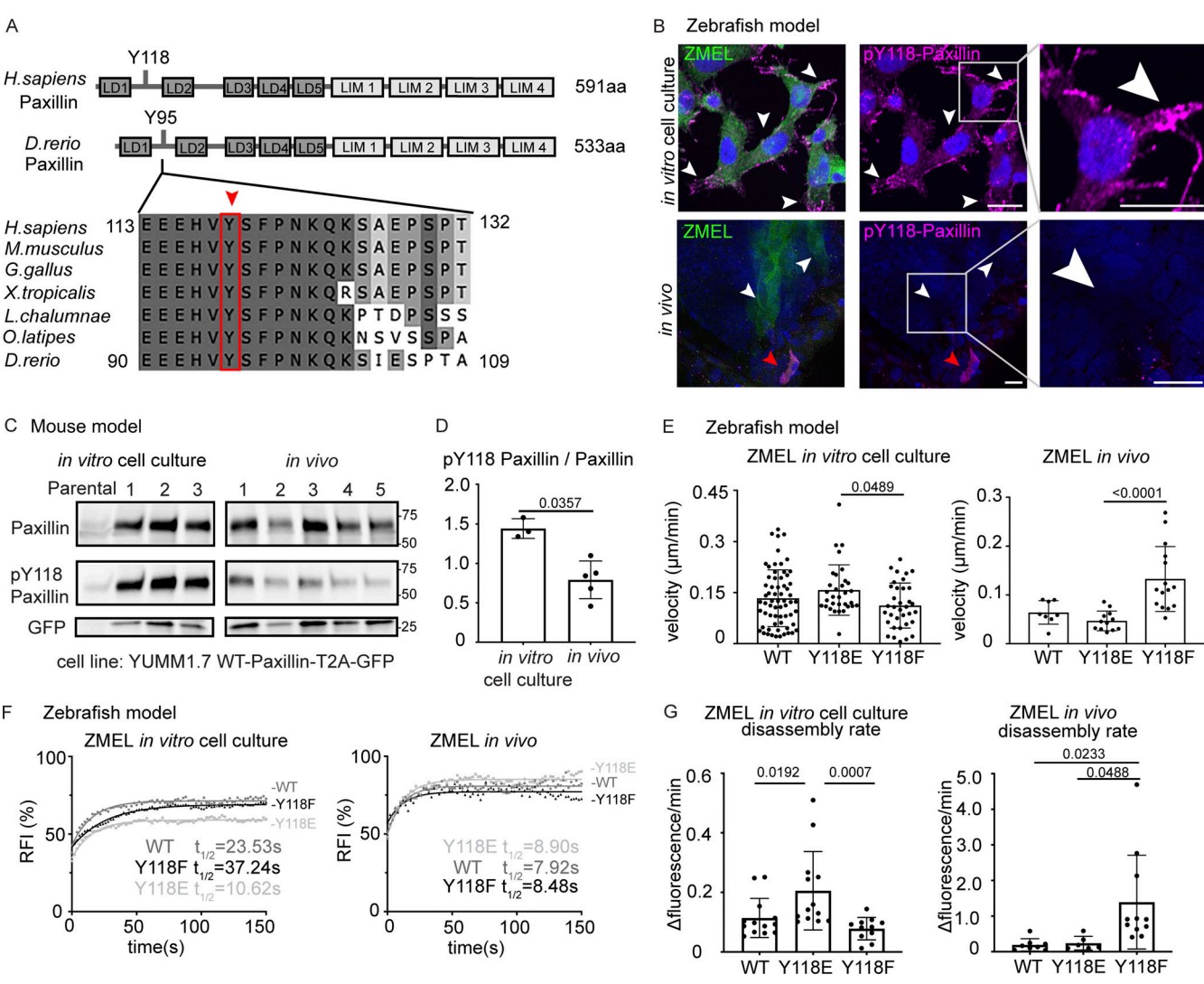

**Figure 3. Paxillin exhibits reduced phosphorylation on Y118 in migrating cancer cells in vivo as compared to in vitro cell culture conditions in both zebrafish and mouse melanoma models. (A)** Schematic of protein structures of human and zebrafish Paxillin (top) and amino acid sequence comparisons of the region encompassing Y118 between zebrafish Paxillin and vertebrate Paxillin (bottom). Red arrowhead and box indicate the conservation of Y118 Paxillin between zebrafish and other vertebrates. **(B)** Top: Endogenous pY118-Paxillin staining (magenta) of ZMEL-GFP (GFP immunostaining, green) plated on 2D in vitro cell culture dishes. White arrowheads mark positive pY118-Paxillin staining. Bottom: Endogenous pY118-Paxillin staining (magenta) of ZMEL-mCherry (mCherry immunostaining, pseudo-colored green, white arrowheads) in larval zebrafish (3 d post-transplantation). Red arrowhead indicates a non-ZMEL cell with positive pY118-Paxillin immunostaining. Zoomed regions reveal pY118-Paxillin immunostaining only. Scale bar is 10 μm. **(C)** Western blot analysis of mouse melanoma YUMM1.7 cells expressing mammalian WT-Paxillin-T2A-GFP plated on the in vitro cell culture dishes (n = 3 dishes) and YUMM1.7 melanoma in vivo tumors (n = 5 tumors). In vitro and in vivo bands are from the same blot—see unmodified Western blot in Fig. S2 D. GFP was used as the loading control and a control for the number of YUMM1.7 cells in mouse tumors. **(D)** Quantification of pY118-Paxillin/total Paxillin protein ratio from C. Non-parametric unpaired t-test. **(E)** Quantification of single cell migration velocity in ZMEL-mCherry cells that exogenously express GFP-tagged zebrafish WT-Paxillin, Y118E-Paxillin, or Y118F-Paxillin in the in vitro cell culture conditions (n = 64 cells for WT, n = 32 cells for Y118E, and n = 35 cells for Y118F) and in vivo (n = 8 cells/3 fish for WT, n = 12 cells/3 fish for Y118E, and n = 15 cells/3 fish for Y118F). Larval zebrafish are imaged 1 d post-transplantation. Non-parametric one-way ANOVA, error bars are mean ± SD. **(F)** Cumulative FRAP recovery curves of WT-Paxillin-EGFP, Y118E-Paxillin-EGFP, or Y118F-Paxillin-EGFP in ZMEL cells in the in vitro cell culture conditions and in vivo after photobleaching. n = 34, 44, and 51 cells for WT, Y118E, Y118F in vitro, and n = 7 cells/6 fish, 6 cells/6 fish, 6 cells/5 fish for WT, Y118E, Y118F in vivo. **(G)** Quantification of Paxillin disassembly rates in the WT, Y118E, Y118F-Paxilllin under in vitro cell culture conditions and WT, Y118E, Y118F-Paxilllin under in vivo conditions. n = 13, 13, and 11 cells for WT, Y118E, Y118F in vitro, and n = 8 cells/7 fish, 6 cells/6 fish, 11 cells/10 fish for WT, Y118E, Y118F in vivo. Error bars are mean ± SD. Non-parametric unpaired t test. Source data are available for this figure: SourceData F3.

skin tissue environment staining positive for pY118-Paxillin (Fig. 3 B, lower panel, red arrowhead), further revealing that the lack of pY118-Paxillin in ZMEL cells in vivo is not due to a technical issue. These findings suggest that phosphorylation of a key residue (Y118) in Paxillin, which is highly enriched and essential for cell migration in vitro, is reduced in vivo.

Paxillin phosphorylation status has been shown to be regulated by stiffness in the environment (Stutchbury et al., 2017). Therefore, we hypothesized that the lack of Y118-Paxillin phosphorylation may be due to the softer environment in vivo compared with the stiffness of glass in culture. To test this hypothesis, we plated ZMEL cells on varying stiffness

environments from 0.5 kPa to glass and found that Y118-Paxillin phosphorylation immunostaining appeared unchanged across the conditions (Fig. S2 C). Thus, the lack of Y118-Paxillin phosphorylation in ZMEL cells in vivo is likely not due to environmental stiffness.

Due to the unexpected finding that Y118-Paxillin does not appear to be phosphorylated in migrating ZMEL cells in vivo, we next wanted to determine whether this phenomenon is also observable in mammalian systems. Thus, we generated melanoma tumors in mice by injecting C57BL/6J mice with YUMM1.7 cells, a highly metastatic murine melanoma cell line, expressing wildtype mammalian Paxillin with a self-cleavable (T2A) GFP tag. We harvested tumors once they reached 1 cm³ and performed Western blot analysis to compare the Y118-Paxillin phosphorylation status of YUMM1.7 cells in vivo to YUMM1.7 cells in the in vitro cell culture conditions. Consistent with our zebrafish results, we found that YUMM1.7 cells in vivo consistently display a significant reduction of phosphorylation of Y118-Paxillin across all tumors we tested (Fig. 3, C and D; and Fig. S2 D). These results, together with our zebrafish results, show that in vivo, migrating cells exhibit significantly reduced levels of phosphorylated Y118-Paxillin. These results suggest that the molecular regulation of focal adhesion components is different in vivo, and thus we began testing the hypothesis that lack of phosphorylation of Y118-Paxillin may facilitate cell migration in vivo.

### Non-phosphorylatable Y118-Paxillin promotes single-cell migration in vivo

To directly test the role of Y118-Paxillin phosphorylation during in vivo cell migration, we performed site-directed mutagenesis to generate phosphomimetic and non-phosphorylatable versions of Y118-Paxillin. We sought to use the zebrafish in vivo system to test the function of these mutations due to the ability to visualize cell migration at high spatial and temporal resolution. We replaced the Y118 residue in zebrafish *paxillin* (which is residue Y95 in zebrafish *paxillin*, but will be henceforth referred to as Y118 for clarity) with a negatively charged glutamic acid (Y118E) to mimic constitutively active tyrosine phosphorylation, and replaced Y118 with phenylalanine (Y118F) to generate a non-phosphorylatable residue, as has been previously described in mammalian cells (Zaidel-Bar et al., 2007b; Petit et al., 2000). We generated new ZMEL lines and first determined the expression levels of each of these constructs and isolated ZMEL populations with robust and ~10-fold overexpression levels of GFP-tagged Y118E or Y118F-Paxillin (Fig. S3 A). We also rederived the WT-Paxillin ZMEL line to use as an internal control (Fig. S3 A). We confirmed that these constructs localize to focal adhesion structures in ZMEL cells in the in vitro cell culture conditions (Fig. S3 B). We then assessed the single-cell migration velocity of ZMEL cells expressing similar levels of GFP-WT/Y118E/Y118F-Paxillin in vitro and in vivo. Under cell culture conditions, similar to what was observed in mammalian cell culture (Petit et al., 2000), ZMEL cells expressing the non-phosphorylatable version of Paxillin (Y118F) exhibited decreased cell migration velocities compared with the phosphomimetic Y118E-Paxillin expressing cells, without affecting directional persistence

(Fig. 3 E, left panel; Fig. S4, A and E). However, when ZMEL cells were transplanted in vivo, ZMEL cells expressing the non-phosphorylatable version of Paxillin (Y118F) exhibited faster migration velocities than both the Y118E-Paxillin expressing cells and the wildtype-Paxillin controls, without affecting directional persistence (Fig. 3 E, right panel; Fig. S4, B and E). Overall, these results reveal the unexpected finding that preventing phosphorylation of Y118-Paxillin has the opposite effect on cell migration in vivo versus in vitro. Non-phosphorylatable Y118-Paxillin exhibits reduced cell migration in vitro compared with the phosphomimetic Y118-Paxillin, but non-phosphorylatable Y118-Paxillin promotes cell migration in vivo.

We next determined how the phosphorylation status of Y118-Paxillin affects focal adhesion dynamics. We first quantified molecular binding dynamics with FRAP analysis as in Fig. 2. We found that while Paxillin exhibited faster molecular binding dynamics in vivo than in vitro, there were no significant differences in molecular binding dynamics between wildtype, Y118E, and Y118F-Paxillin in vivo (Fig. 3 F and Fig. S4 D). This was an expected result as previous work had shown that mutations affecting the phosphorylation state of focal adhesion proteins did not affect mobility kinetics (Stutchbury et al., 2017). In contrast, in cell culture, we observed differences in molecular binding dynamics in cells expressing WT, Y118E, and Y118F-Paxillin, with Y118E exhibiting faster $t_{1/2}$ values and Y118F exhibiting slower $t_{1/2}$ values (Fig. 3 F and Fig. S4 D). These results again reveal differences in regulation between in vitro and in vivo conditions.

We then sought to quantify focal adhesion lifetimes in ZMEL cells expressing Y118-Paxillin mutants. Due to signal-to-noise challenges of the Paxillin-GFP constructs in vivo, we were unable to detect the initial stages of focal adhesion formation in ZMEL cells expressing these constructs and were thus unable to quantify accurate lifetimes of the structures. However, we were able to quantify focal adhesion disassembly rates by determining the maximum fluorescence intensity of Paxillin-positive structures and analyzing the decrease in fluorescence over time. When we analyzed disassembly rates in ZMEL cells expressing WT, Y118E, and Y118F-Paxillin under in vitro cell culture conditions, we found that ZMEL cells expressing Y118E-Paxillin exhibited faster focal adhesion disassembly rates than Y118F-Paxillin expressing ZMEL cells (Fig. 3 G). These results are consistent with the increased migration rate also observed in Y118E-Paxillin expressing ZMEL cells in culture compared with Y118F-Paxillin expressing ZMEL cells. We also found that ZMEL cells expressing the non-phosphorylatable Y118F-Paxillin in vivo exhibited faster disassembly rates than ZMEL cells expressing the phosphomimetic or wildtype Paxillin (Fig. 3 G). These results suggest that in vivo, cells expressing non-phosphorylatable Y118-Paxillin exhibit increased cell migration velocities by increasing the turnover of focal adhesion structures.

### Paxillin phosphoregulation is conserved in macrophages in vivo

Due to the unexpected discovery that the non-phosphorylatable Y118F mutant of Paxillin promotes ZMEL cell migration in vivo, we next asked whether other migrating cells in vivo are

Figure 4. **Macrophages expressing non-phosphorylatable Y118F-Paxillin exhibit increased motility in vivo. (A)** Endogenous pY118 Paxillin immunostaining (magenta) of macrophages (green, white arrowheads) in Tg(mpeg:Lifeact-GFP)$^{zj506}$ larval zebrafish. Red arrowhead marks positive pY118 Paxillin immunostaining of a non-macrophage cell. Zoomed region of macrophage lacking pY118-Paxillin immunostaining. **(B)** Schematic of zebrafish tail wound transection area and macrophage imaging area for directed cell migration. **(C)** Still images from zebrafish macrophage tracking timelapse videos in 3 dpf Tg(mpeg:WT-zebrafish Paxillin- EGFP)$^{zj503}$, Tg(mpeg:zebrafish Y118E-Paxillin- EGFP)$^{zj504}$, and Tg(mpeg:zebrafish Y118F-Paxillin-EGFP)$^{zj505}$ larvae at timepoint 0 and 10 min. Dotted lines indicate wound sites and arrows show the direction of migration. See also Video 7. Scale bar is 10 μm. **(D)** Quantification of macrophage migration velocities toward the wound in vivo. Non-parametric one-way ANOVA, error bars are mean ± SD. n = 38 cells/6 fish for WT, n = 20 cells/6 fish for Y118E and n = 24 cells/10 fish for Y118F. **(E)** Cell tracking of macrophage migration trajectories toward the wound in vivo, migration starting points are normalized to 0 in both x and y axes, wound sites are normalized to the positive x axis (n = 38 cells/6 fish for WT, n = 20 cells/6 fish for Y118E and n = 24 cells/10 fish for Y118F). Arrows show the direction of migration toward the wound.

similarly affected by Y118-Paxillin phosphorylation status. We tested macrophages, which are highly migratory immune cells that have also been shown to form Paxillin-positive punctae during migration in vivo (Barros-Becker et al., 2017). Similar to ZMEL cells, migrating macrophages in larval zebrafish do not have detectable pY118-Paxillin staining (Fig. 4 A). We then generated stable transgenic zebrafish strains in which macrophages either expressed wildtype zebrafish *paxillin* Tg(mpeg:WT-Paxillin-EGFP) $^{zj503}$, the phosphomimetic Tg(mpeg:Y118E-Paxillin-EGFP)$^{zj504}$, or the non-phosphorylatable Tg(mpeg: Y118F-Paxillin-EGFP)$^{zj505}$ versions of zebrafish *paxillin*, and we crossed these strains to a macrophage

reporter line Tg(mfap4:tdTomato-CAAX)$^{xt6}$ for better macrophage visualization due to the low expression of the Paxillin-EGFP constructs to compare macrophage migration velocities during directed cell migration. We created a wound in the larval tail and imaged macrophage migration toward the wound within 4 h with live-cell microscopy (Fig. 4 B). Similar to what we observed with ZMEL cells in vivo (Fig. 3 E, right panel), macrophages expressing the non-phosphorylatable Y118-Paxillin (Y118F) showed increased migration velocity as compared with macrophages expressing the phosphomimetic Y118E-Paxillin or wildtype-Paxillin controls (Fig. 4, C–E; Fig. S4 C; and Video 7). These results demonstrate that

in two distinct cell types, expressing the non-phosphorylatable version of Y118-Paxillin enhances cell migration in vivo.

## FAK is downregulated and CRKII-DOCK180/RacGEF exhibits increased interaction with unphosphorylated Y118-Paxillin in vivo

Next, we sought to understand the intrinsic regulation of Y118-Paxillin that leads to reduced phosphorylation in vivo. We used mouse YUMM1.7 melanoma tumors to evaluate both upstream and downstream focal adhesion signaling. We first evaluated the well-known upstream kinase, FAK, which is known to phosphorylate Paxillin on Y118 (Fig. 5 A). FAK kinase activity is activated by autophosphorylation at tyrosine 397 (Y397) upon integrin-mediated cell adhesion (Mitra et al., 2005; Schaller and Parsons, 1995; Bellis et al., 1995). We first performed Western blot analysis using a pY397-FAK antibody to compare the FAK kinase activity of YUMM1.7 cells in vivo to YUMM1.7 cells that are cultured in vitro. Surprisingly, FAK activity (as measured by pY397-FAK levels over total FAK levels) remains unchanged (Fig. 5, B and C; and Fig. S5 A). However, the total FAK protein abundance is significantly decreased in vivo as compared with in vitro cell culture conditions (Fig. 5, B and D; and Fig. S5 A). These results suggest that the lack of Y118-Paxillin phosphorylation in vivo may be due to a significant reduction in the expression of the upstream kinase, FAK. To test this hypothesis, we next wanted to functionally test whether FAK was capable of phosphorylating Paxillin in vivo. Given that FAK expression levels are already low in cells in vivo (Fig. 5, B and D), we focused on experiments overexpressing FAK. We overexpressed mammalian GFP-FAK in mouse YUMM1.7 melanoma cells and determined whether we could detect a corresponding increase in pY118-Paxillin in cells in vivo. We performed Western blot analysis on GFP-FAK-expressing YUMM1.7 cells and first confirmed the overexpression of FAK (Fig. S5 B). In the GFP-FAK-overexpression tumors, we found an increase in pY118-Paxillin levels compared with control GFP-expressing YUMM1.7 tumors (Fig. 5, E and F). Together these results suggest that FAK is capable of phosphorylating Paxillin on Y118 in vivo, but that the low levels of pY118-Paxillin observed in cells in vivo are likely due to low levels of FAK.

We next aimed to dissect the underlying molecular players that promote cell migration in cells expressing the non-phosphorylatable version of Y118-Paxillin in vivo. We generated mouse YUMM1.7 melanoma tumors either expressing the wildtype, the phosphomimetic (Y118E), or the non-phosphorylatable (Y118F) version of mammalian Paxillin and fused the Paxillin mutants to a self-cleavable GFP (T2A-GFP); thus, GFP localization is uncoupled from Paxillin localization in these cells. Then, we sought to investigate interactions of Paxillin with known binding partners to determine if perturbation of the Y118 residue influences these interactions. We focused on Vinculin and CRKII, which have both been shown to associate with Paxillin in response to Paxillin tyrosine phosphorylation in in vitro cell culture conditions (Birge et al., 1993; Schaller and Parsons, 1995; Petit et al., 2000; Pasapera et al., 2010; Case et al., 2015). We first tested interactions between Paxillin and Vinculin by performing immunoprecipitations of Paxillin and assaying

for Vinculin interactions. We did not observe any differences in Paxillin–Vinculin interactions in YUMM1.7 cells expressing Y118-Paxillin mutations in vivo (Fig. S5 C). However, strikingly, when we assayed for Paxillin–CRKII interactions, we found that expression of the non-phosphorylatable Y118F-Paxillin leads to significantly enhanced Paxillin–CRKII interactions in vivo, with no observable differences in vitro (Fig. 5, G–J).

Given the critical role of CRKII in downstream cell migration (Lamorte et al., 2003), we further evaluated proteins that function downstream of CRKII to more directly evaluate the effects of non-phosphorylatable Paxillin on cell migration. Paxillin–CRKII complexes can interact with a number of downstream proteins, and we evaluated proteins that are recruited to focal adhesions to regulate cell migration. We specifically focused on downstream effectors of the Rho and Ras families of GTPases given their well-characterized functions in focal adhesion formation and subsequent cell migration (Bar-Sagi and Hall, 2000; Ohba et al., 2001; Ridley, 2001). We analyzed two guanine exchange factors (GEFs) that activate two distinct GTPases—DOCK180 (a Rac GEF) and C3G (a Ras GEF; Kiyokawa et al., 1998a; Tanaka et al., 1994). Given that both of these GEFs also function at the leading edge of migrating cells (Ichiba et al., 1997; Santy et al., 2005), we sought to specifically test DOCK180/RacGEF or C3G/RasGEF interactions with Paxillin at focal adhesions, and thus we took advantage of biochemical approaches to assay for protein interactions, rather than knocking down GEF function. Using mouse melanoma tumors expressing either wildtype, phosphomimetic (Y118E), or non-phosphorylatable (Y118F) versions of mammalian Paxillin, we immunoprecipitated Paxillin and found there is no difference in the level of C3G interactions with Paxillin when cells are expressing wildtype, Y118E, or Y118F-Paxillin in vivo or in vitro (Fig. S5, D, E, G, and H). We also assayed ERK activity, which is downstream of C3G activity, and found no difference in the phosphorylated active state of ERK across all conditions in vivo or in vitro (Fig. S5, D, F, G, and I). In contrast, we observed a dramatic enrichment of DOCK180/RacGEF interacting with non-phosphorylatable Y118F-Paxillin in vivo compared with cells expressing the phosphomimetic or wildtype Paxillin (Fig. 5 K). Together, these results suggest that non-phosphorylatable Y118F-Paxillin exhibits increased cell migration in vivo through increased activation of the CRKII-DOCK180/RacGEF signaling pathway.

Our results so far compare CRKII-DOCK180/RacGEF signaling activation between cells expressing Y118-Paxillin phosphomutants. We next wanted to compare the pathway activation status between migrating cells in vivo versus in vitro in the absence of any mutant (Y118E or Y118F) Paxillin overexpression constructs. We already demonstrated that Paxillin has reduced phosphorylation at Y118 in migrating cells in vivo (Fig. 3, B–D), thus we hypothesized that Paxillin will exhibit increased interaction with CRKII and DOCK180/RacGEF in cells in vivo compared with in vitro. Using mouse melanoma tumors expressing wildtype Paxillin, which already exhibit low levels of Paxillin Y118 phosphorylation in vivo (Fig. 3, C and D), indeed we found that a significantly higher amount of CRKII and DOCK180/RacGEF interacted with wildtype Paxillin in cells in vivo versus

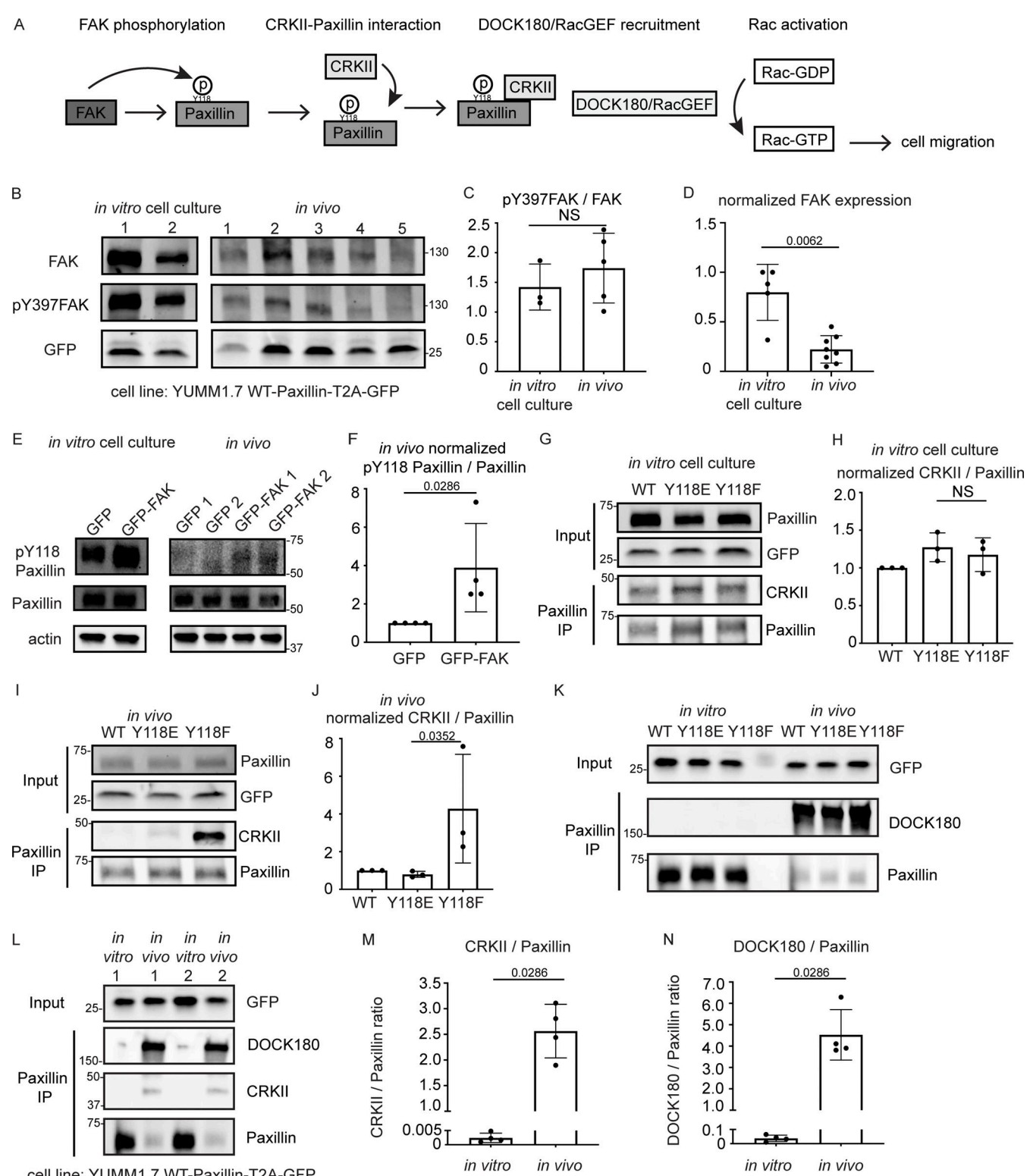

Figure 5. **FAK is downregulated and CRKII-DOCK180/RacGEF exhibits increased interaction with unphosphorylated Y118-Paxillin in vivo compared to in vitro. (A)** Schematic of in vitro Paxillin regulation from cell culture studies. Following integrin activation, a tyrosine kinase, FAK, phosphorylates Paxillin. Phosphorylated Paxillin then recruits the adaptor protein CRKII and the Paxillin/CRKII complex further recruits DOCK180/RacGEF, thereby activating downstream Rac-dependent pathways, inducing cell migration. **(B)** Western blot analysis of FAK levels (FAK) and FAK activation (pY397-FAK) in YUMM1.7 cells expressing mammalian WT-Paxillin-T2A-GFP in culture and YUMM1.7 tumors in vivo. In vitro and in vivo bands are from the same blot. Unmodified Western blot is in Fig. S5 A. GFP was used as the loading control and as a control for the number of YUMM1.7 cells in mouse tumors. **(C and D)** Quantification of the pY397-FAK/total FAK ratio (C) and total normalized FAK to GFP expression (D) in the in vitro cell culture and in vivo conditions. $n$ = 3 dishes, 5 tumors for C, $n$ = 5 dishes, 8 tumors for D. Error bars are mean ± SD. Non-parametric unpaired $t$ test. **(E)** Western blot analysis of pY118-Paxillin levels in YUMM1.7 cells

overexpressing GFP-FAK in vitro and in vivo. Actin is used as a loading control. **(F)** Quantification of pY118-Paxillin/Paxillin levels in E. GFP control tumors are normalized to 1. $n$ = 4 technical replicates. Error bars are mean ± SD. Non-parametric unpaired $t$ test. **(G–J)** Co-immunoprecipitation analyses of CRKII and Paxillin in YUMM1.7 cell lines that exogenously express mammalian wildtype, Y118E and Y118F Paxillin in vitro (G and H) and in in vivo tumors (I and J). **(H and J)** Quantification of CRKII/Paxillin ratio from G and I, bands from cells expressing wildtype Paxillin are normalized to 1 both in vitro and in vivo. $n$ = 3 technical replicates. Non-parametric one-way ANOVA, error bars are mean ± SD. **(K)** Coimmunoprecipitation analyses of DOCK180/RacGEF and Paxillin in YUMM1.7 cell lines that exogenously express mammalian wildtype, Y118E and Y118F Paxillin in vitro and in in vivo tumors. **(L–N)** Coimmunoprecipitation analyses of CRKII and DOCK180/RacGEF to Paxillin in YUMM1.7 cell lines that exogenously express wildtype Paxillin in in vitro and in in vivo tumors. **(M)** Quantification of CRKII/Paxillin levels in L. $n$ = 4 tumors. **(N)** Quantification of DOCK180/Paxillin levels in L. $n$ = 4 tumors. Error bars are mean ± SD. Non-parametric unpaired $t$ test. Source data are available for this figure: SourceData F5.

cells in culture (Fig. 5, L–N). Taken together, these results suggest a mechanism (Fig. 6) for how Paxillin regulates cell migration in vitro versus in vivo: in cultured cells, Y118-Paxillin is phosphorylated, leading to efficient cell migration. Lack of Y118-Paxillin phosphorylation in cells in culture leads to reduced cell migration compared to expression of the phosphomimetic. However, in vivo, Paxillin exhibits reduced phosphorylation at Y118 in migrating cells, likely due to lower in vivo levels of the upstream kinase, FAK. Lack of phosphorylation at Y118 on Paxillin, or the expression of non-phosphorylatable Y118F-Paxillin, leads to increased focal adhesion disassembly rate,

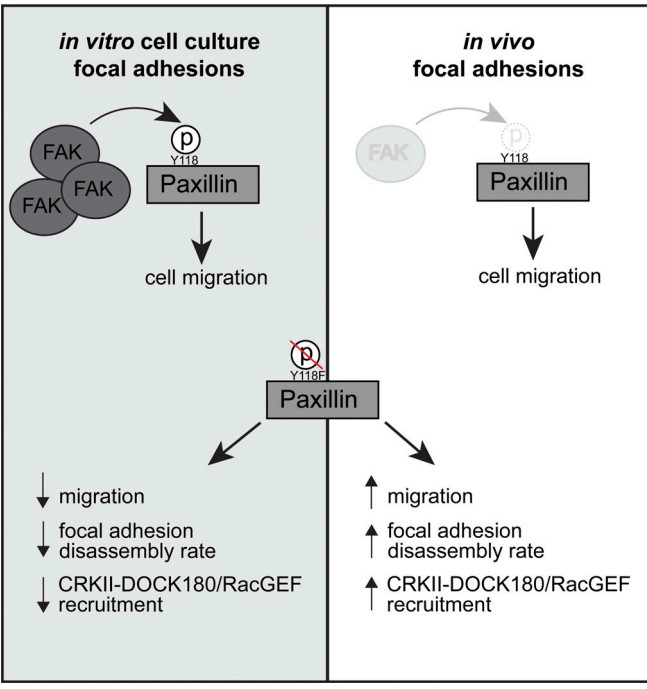

Figure 6. **Working model for how Y118-Paxillin phosphorylation status regulates cell migration in the in vitro cell culture and in vivo conditions.** Top: Under in vitro cell culture conditions, FAK phosphorylates Paxillin on Y118, leading to high levels of Y118-Paxillin phosphorylation in migrating cells. In migrating cells in vivo, FAK levels are low, and Y118-Paxillin lacks phosphorylation. Bottom: Expression of the non-phosphorylatable Y118F-Paxillin leads to reduced cell migration in vitro compared with cells expressing the Y118E-Paxillin phosphomimetic, likely through reduced focal adhesion disassembly rates and reduced CRKII-DOCK180/RacGEF recruitment to Paxillin-positive focal adhesions. However, in vivo, cells expressing the non-phosphorylatable Y118F-Paxillin exhibit increased cell migration, likely through increased focal adhesion disassembly rates, and increased recruitment of CRKII-DOCK180/RacGEF to Paxillin-positive focal adhesions.

increased CRKII and DOCK180/RacGEF recruitment, and increased cell migration in vivo.

## Discussion

Understanding focal adhesion regulation in 3D environments and animal models is an emerging field in cell migration. The molecular components that comprise a focal adhesion and the intrinsic regulation of these components is still entirely unknown during single-cell migration in vivo. In this work, we developed a zebrafish syngeneic transplantation model where focal adhesion structures can be efficiently visualized with high resolution during single-cell migration in vivo. Taking advantage of this in vivo approach, we performed a direct comparison of a core focal adhesion protein, Paxillin, between in vivo and the in vitro cell culture conditions (Fig. 6). Using zebrafish and mouse models, we found that Y118-Paxillin, a key residue that has been shown to be phosphorylated in vitro, has significantly reduced phosphorylation in vivo, and this lack of phosphorylation in vivo is likely due to a significant reduction of the upstream kinase, FAK. Furthermore, preventing phosphorylation of Y118-Paxillin inhibits cell migration in vitro compared with the expression of the Y118E-Paxillin phosphomimetic, but preventing phosphorylation of Y118-Paxillin promotes cell migration in vivo in both migrating tumor cells and macrophages, and these differences correlate with the differential cellular interactions of CRKII and DOCK180/RacGEF with Paxillin. Interestingly, preventing phosphorylation of Y118-Paxillin also increased the rate of focal adhesion disassembly in vivo, despite previous studies indicating that lack of Y118-Paxillin phosphorylation inhibits focal adhesion disassembly in cell culture (Zaidel-Bar et al., 2007a). Altogether, our results show that Paxillin phosphorylation status at Y118 leads to opposite phenotypes on cell migration in vivo versus in vitro cell culture conditions. The in vivo nature of these experiments makes it difficult to discern the specific aspects of the environment that dictate these differences, in part due to the length of time required to generate transgenic or mutant animals, as well as the fact that mutations affecting the environment would require whole-body manipulations that could lead to indirect effects. Additionally, focal adhesions are likely to be regulated by a multitude of environmental factors and not a single variable. Despite these challenges, our findings still represent the first analysis of focal adhesion component dynamics and regulation during single-cell migration in vivo and identify an altogether new molecular mechanism for how Paxillin regulates focal adhesion-based migration, a critical step in moving the cell migration field forward.

Overexpression of Paxillin has been reported in a number of different human cancer types (Sobkowicz et al., 2017; Deakin et al., 2012; Yang et al., 2010; Salgia et al., 1999; Mackinnon et al., 2011). However, there are far fewer studies investigating Paxillin phosphorylation in cancer progression. Previous research indicates that Y118-Paxillin phosphorylation levels are inversely correlated with cancer metastasis in breast cancer patient tissue samples (Madan et al., 2006), suggesting that, similar to our findings, Y118-Paxillin phosphorylation does not correlate with an invasive phenotype. However, contrary to these findings, it has also been shown that Y118-Paxillin phosphorylation correlates with advanced human osteosarcoma metastatic stages, with highly metastatic osteosarcoma cell lines expressing high levels of pY118-Paxillin and lowly metastatic cell lines expressing low levels of pY118-Paxillin (Azuma et al., 2005). However, these experiments were performed in cell culture systems; thus, it is unclear how Y118-Paxillin is regulated in vivo in this system. Our melanoma work in cell culture systems is consistent with these previous in vitro observations, and in vivo, our work suggests that migrating cancer cells exhibit low levels of pY118-Paxillin, highlighting functional differences of Y118-Paxillin phosphorylation status in cancer cell migration in vivo versus in vitro. The dramatic increase in Paxillin–CRKII-DOCK180/RacGEF interactions in Y118F-Paxillin-expressing cells in the in vivo mouse tumors was surprising, but it provides mechanistic support underlying the increased cell motility observed in Y118F-Paxillin-expressing cells in zebrafish, although other downstream pathways are also likely to be involved in this process. CRKII has been known to directly bind Paxillin at YXXP motifs, including the phosphorylated Y118 residue of Paxillin in cultured cells (Birge et al., 1993; Schaller and Parsons, 1995). However, our results suggest that CRKII may bind another site on Paxillin in vivo and that this interaction is enhanced in the absence of phosphorylation at Y118-Paxillin. Consistent with this hypothesis, previous data have suggested that CRKII can bind Paxillin at non-YXXP motifs (Takino et al., 2003), depending on the phosphorylation status of CRKII.

Mutating Y118-Paxillin also affected the disassembly rate of Paxillin-positive focal adhesion structures. Expression of the non-phosphorylated Y118F-Paxillin led to increased rates of focal adhesion disassembly in vivo, without changing the mobility kinetics of the Paxillin molecules in and out of the structure. These results are consistent with the increased cell migration velocities observed in Y118F-Paxillin-expressing cells in vivo, as increasing the focal adhesion turnover has been shown to lead to increased cell migration (Meenderink et al., 2010; Nagano et al., 2012; Webb et al., 2004).

In this work, we exclusively tested the function of Y118-Paxillin in migrating cells in vivo. However, the phosphorylation of Y118-Paxillin is concomitant with phosphorylation at another tyrosine residue, Y31 (Schaller and Parsons, 1995). These two tyrosine residues have been largely investigated together (Tsubouchi et al., 2002; Zaidel-Bar et al., 2007b; Petit et al., 2000) and have been shown to play a critical role in focal adhesion turnover for efficient cell migration in cell culture (Zaidel-Bar et al., 2007a). We also evaluated the status of Y31-Paxillin in our studies and found that phosphorylation of Y31-Paxillin was similarly downregulated in cells in vivo compared with cell culture (Fig. S5, J and K), suggesting that these two

residues might function synergistically during cell migration in vivo. However, our work also suggests that preventing phosphorylation of Y118-Paxillin alone can regulate focal adhesion dynamics during cell migration in vivo, and future work is required to explore the function of Y31-Paxillin, either alone or together with Y118-Paxillin.

The upstream kinase of Paxillin, FAK, is also shown to be highly expressed and activated in many types of cancers (Miyazaki et al., 2003; Itoh et al., 2004; Cance et al., 2000; Madan et al., 2006), and has been used as a therapeutic target for cancer treatment, including pancreatic cancer and non-small-cell lung carcinoma (Gerber et al., 2020; Parsons et al., 2008). The mechanism of FAK inhibition in cancer progression is complex because FAK has been shown to play a critical role at focal adhesions in cell culture and has also been shown to play signaling roles for cell survival and proliferation (Sulzmaier et al., 2014). Knowledge of FAK inhibition in regulating cancer cell migration or cancer metastasis is largely based on in vitro cell culture studies where FAK inhibitors alter the dynamics and formation of focal adhesion structures (Stutchbury et al., 2017; Chan et al., 2009) and downstream signals (Sieg et al., 2000; Meng et al., 2009), leading to reduced cell migration. However, it is unclear whether FAK regulates cell migration in vivo by the same mechanism that is described in cell culture systems or whether focal adhesion structures are perturbed by FAK inhibitors in vivo, as a major limitation is the difficulty in visualizing transient and subcellular focal adhesion structures in vivo. In this study, we developed an animal system where focal adhesion structures can be readily visualized in vivo. We also found that FAK levels are reduced in cancer cells in vivo compared with cell culture systems and that disrupting a key site for FAK phosphorylation on Paxillin leads to enhanced cancer cell migration in vivo. While inhibiting FAK function and dissecting focal adhesion dynamics would be ideal, inhibiting FAK activity has already been shown to inhibit cell migration in vivo (Sulzmaier et al., 2014; Megison et al., 2013; Liu et al., 2007; Cabrita et al., 2011), thus this result precludes the analysis of focal adhesion dynamics. Future work manipulating FAK function spatially and temporally in vivo is required to further test this process. Together, our results suggest that FAK regulation of cancer progression in vivo may be mediated in large part through signaling mechanisms that regulate cell survival or proliferation, and perhaps less so at focal adhesions.

Our work focuses on a single core member of focal adhesions—Paxillin. It is unknown how other focal adhesion proteins, particularly proteins from different functional layers within the focal adhesion architecture (Kanchanawong et al., 2010), are dynamically regulated in vivo and whether their regulation is distinct from the in vitro cell culture model. It is possible that in vivo focal adhesions have a different multilaminar organization as compared with in vitro cell culture studies. Future work will be required to investigate these and other fascinating questions.

## Materials and methods
### Experimental models and subject details
#### Zebrafish (Danio rerio)
Zebrafish were raised in the Centralized Zebrafish Animal Resource (CZAR) at the University of Utah and experiments were

approved by the Institutional Animal Care and Use Committee. Previously described transgenic zebrafish lines used were: *Tg(mitfa:BRAFV600E);p53⁻/⁻;mitfa⁻/⁻*, *Tg(mfap4:tdTomato-CAAX)ˣᵗ⁶*; and *Tg(mpeg1:lifeact-EGFP)ᶻʲ⁵⁰⁶*.

### Mouse (Mus musculus)
All mouse work was performed by Preclinical Research Shared Resource at Huntsman Cancer Institute at the University of Utah, following IACUC and AAALAS guidelines. C57BL/6J mice were purchased from Jackson Laboratory (000664).

## Methods details
### Cloning of Y118 Paxillin mutations
Zebrafish Paxillin Y118 phosphorylation mutants were constructed as previously described (Zaidel-Bar et al., 2007b) with mEGFP tags to the N-terminus. Zebrafish paxillin Y95 is the ortholog of human paxillin Y118, so zebrafish paxillin 95 was replaced with glutamic acid (phosphomimetic) or phenylalanine (non-phosphorylatable). *pCS2+Paxillin-mKate* (plasmid #105974; Addgene) was used as the template for amplifying zebrafish paxillin-a (pxna). 1-2E/2F fragment was amplified using primer1 and primer2E/2F (Tm = 71°C) and gel-purified, 3 + 4 fragment was amplified using primer3 and primer4 (Tm = 59°C) and gel-purified. Then the 1-2E/2F and 3 + 4 fragments were combined for overlap PCR using primer1 and primer4' (Tm = 55°C→70°C). *pCS2+Paxillin-mKate* was also used to amplify wildtype pxna by using primer1 and primer4' (Tm = 70°C). PCR products were gel-purified and digested with BglII and EcoRI, and finally ligated with mEGFP-C1 (plasmid #54759; Addgene) to make the final plasmids *mEGFP-WT-pxna*, *mEGFP-Y118E-pxna*, and *mEGFP-Y118F-pxna*.

Primer1: 5'-GAAGATCTATGGACGATTTAGATGCTCTTCTCG CGG-3'. Primer2E: 5'-CTGTTTGTTGGGGAAACTCTCGGCGTGCT CTTC-3'. Primer2F: 5'-CTGTTTGTTGGGGAAACTGAAGGCGTG CTCTTC-3'. Primer3: 5'-AGTTTCCCCAACAAACAG-3'. Primer4: 5'-GCTGAAGAGCTTGACGAAG-3'. Primer4': 5'-CGAATTCCTAG CTGAAGAGCTTGACGAAG-3'.

Human Paxillin Y118 phosphorylation mutants were cloned similar to zebrafish constructs. Briefly, *pmCherry Paxillin* (plasmid #50526; Addgene) was used as the template. 1-2E/2F fragment was amplified using primer1-h and primer2E-h/2F-h (Tm = 72°C), and 3 + 4 fragment was amplified using primer3-h and primer4-h (Tm = 60°C). Then the 1-2E/2F and 3 + 4 fragments were combined for overlap PCR using primer1-h and primer4'-h (Tm = 56°C→71°C). Wildtype human Paxillin was amplified directly by using primer1-h and primer4'-h (Tm = 71°C). PCR products were gel-purified and then continued with Gibson cloning.

Primer1-h: 5'-GCCACCATGGACGACCTCGACGCCC-3'. Primer2E-h: 5'-GCTTGTTGGGGAAGCTCTCGACGTGCTCCTC-3'. Primer2F-h: 5'-GCTTGTTGGGGAAGCTGAAGACGTGCTCCTC-3'. Primer3-h: 5'-AGCTTCCCCAACAAGC-3'. Primer4-h: 5'-CTAGCAGAAGAGCTTGA GGAAG-3'. Primer4'-h: 5'-CTAGCAGAAGAGCTTGAGGAAGCAGTT CTG-3'.

Human Paxillin fragments were then Gibson assembled with a T2A-GFP fragment into a modified *pLKO.1* plasmid backbone with accessible multiple cloning sites. Briefly, Paxillin fragments were amplified from previously gel recycled products using PAX-F and PAX-R primers, and T2A-GFP fragment was amplified by T2A-F and GFP-R primers using *plenti-CMV-mCherry-T2A-GFP* (plasmid #109427; Addgene) as a template. Two fragments were cloned from pLKO.1 backbone, which we refer to as A1-A2 and B1-B2 fragments by using the A1-A2 and B1-B2 primer sets listed below. Finally, Paxillin fragments were combined with T2A-GFP, A1-A2, and B1-B2 fragments for Gibson assembly using NEBuilder HiFi DNA Assembly Master Mix (NEB E2621S) to generate *pLKO.1-WT-Paxillin-T2A-GFP*; *pLKO.1-Y118E-Paxillin-T2A-GFP*; and *pLKO.1-Y118F-Paxillin-T2A-GFP* constructs. NEB High Fidelity PCR Master Mix with HF Buffer (M0531S) was used for all PCR reactions, and sequences were verified with Sanger sequencing.

**Gibson assembly primers.** A1: 5'-GTCGAGGTCGTCCATGGT AAGCTCCGGTGACGTC-3'. PAX-F: 5'-TCACCGGAGCTTACCATG GACGACCTCGACGCCCT-3'. PAX-R: 5'-GATCTGCACCGGGGCAG AAGAGCTTGAGGAAGCAGT-3'. T2A-F: 5'-TCCTCAAGCTCTTCT GCCCCGGTGCAGATCTCGA-3'. GFP-R: 5'-ACAGATATCCGTACG TTATTTATATAATTCATCCATACCGAGAG-3'. B2: 5'-GAATTAT ATAAATAACGTACGGATATCTGTACAAGTAACGCCC-3'. B1: 5'-CTTTATCCGCCTCCATCCAGTCTATTAATTGTTGCC-3'. A2: 5'-AATTAATAGACTGGATGGAGGCGGATAAAG-3'.

### Cloning of FAK overexpression constructs
Mouse FAK fragments were then Gibson assembled into a modified *pLKO.1* plasmid backbone with accessible multiple cloning sites. Briefly, GFP-FAK and GFP control fragments were amplified from GFP-FAK WT (plasmid #186148; Addgene) using GFP-FAK-F, GFP-FAK-R and GFP-F, GFP-R primer sets. Two fragments were cloned from pLKO.1 backbone, which we refer to as vector1 and vector2 fragments, by using the vector1-F, vector1-R, and vector2-F (or vector2-GFP-F), vector2-R primer sets listed below. Finally, GFP-FAK and GFP control fragments were combined with vector1 and vector2 fragments for Gibson assembly using NEBuilder HiFi DNA Assembly Master Mix (NEB E2621S) to generate *pLKO.1-GFP-FAK* and *pLKO.1-GFP* constructs. NEB High Fidelity PCR Master Mix with HF Buffer (M0531S) was used for all PCR reactions and sequences were verified with Sanger sequencing.

GFP-FAK-F: 5'-GGGGGATCCGGCCACCATGGTGAGCAAGGG C-3'. GFP-FAK-R: 5'-TTGTACAGATATCTCAGTGTGGCCGTGTC TGC-3'. GFP-F: 5'-GGGGGATCCGGCCACCATGGTGAGCAAGGG C-3'. GFP-R: 5'-TTGTACAGATATCTCACTTGTACAGCTCGTCCA TGCC-3'. Vector1-F: 5'-GGTGCCTCACTGATTAAGCATTGG-3'. Vector1-R: 5'-CCATGGTGGCCGGATCCCCCTGGGG-3'. Vector2-F: 5'-CGGCCACACTGAGATATCTGTACAAGTAACGCCCGC-3'. Vector2-GFP-F: 5'-GAGCTGTACAAGTGAGATATCTGTACAAGTAAC GCCCGC-3'. Vector2-R: 5'-ACCAATGCTTAATCAGTGAGGCACC-3'.

### Zebrafish line generation
All constructs were generated using Gateway Cloning technology (Thermo Fisher Scientific) based on the manufacturer's instructions. To generate Paxillin-expressing melanoma Mini-CoopR fish, zebrafish paxillin-a (*pxna*) was cloned from the *pCS2+Paxillin-mKate* construct (plasmid #105974; Addgene) using the primers listed below (pME-pxna Forward and pME-pxna Reverse). The *pxna* fragment was then cloned into the gateway

donor vector *pDONR221(pME)* to generate *pME-pxna* by using BP clonase (11789020; Thermo Fisher Scientific). Finally, *pME-pxna* was combined with *p5E-mitfa2.1* (plasmid #81234; Addgene), p3E-EGFP (gift from Rodney Stewart, University of Utah, Salt Lake City, UT), and the MiniCoopR destination vector (Ceol et al., 2011) to generate the *pDest-MiniCoopR mitfa2.1:pxna-EGFP* plasmid using LR clonase (11791020; Thermo Fisher Scientific).

**Primers.** pME-pxna Forward: 5′-GGGGACAAGTTTGTACAA AAAAGCAGGCTTCGCCACCATGGACGATTTAGATGCTCTTCTC-3′. pME-pxna Reverse: 5′-GGGGACCACTTTGTACAAGAAAGC TGGGTAGCTGAAGAGCTTGACGAAGC-3′.

Melanoma-bearing MiniCoopR fish were generated by single-cell injection of *pDest-MiniCoopR mitfa2.1-pxna-EGFP* or *pDest-MiniCoopR mitfa2.1-mCherry* plasmid into embryos from *Tg(mitfa:BRAFV600E);p53⁻/⁻;mitfa⁻/⁻* fish with Tol2 transposase RNA as previously described (Ceol et al., 2011).

Collagen reporter fish were generated by single-cell injection of *krt19:col1a2-GFP* plasmid (gift from Paul Martin) into wildtype zebrafish embryos.

To generate zebrafish lines expressing Paxillin mutants in macrophages—*Tg(mpeg1:WT-Paxillin-EGFP)*, *Tg(mpeg1:Y118E-Paxillin-EGFP)*, *Tg(mpeg1:Y118F-Paxillin-EGFP)*—WT-pxna, Y118E-pxna, and Y118F-pxna fragments were amplified using *WT-pxna*, *pxna-Y118E*, and *Y118F-pxna* plasmids (described above) as templates, and pME-pxna Forward/Reverse as primers. These fragments were Gateway cloned into the pME vector to generate *pME-WT-pxna*, *pME-Y118E-pxna*, and *pME-Y118F-pxna* by using BP clonase (11789020; Thermo Fisher Scientific). Then, *pME-WT/Y118E/Y118F-pxna* were combined with *p5E-mpeg1* (Roh-Johnson et al., 2017), *p3E-EGFP-pA* (gift from Rodney Stewart), and *pDestpBHR4R3* (gift from Susan Brockerhoff) to generate *mpeg1:WT/Y118E/Y118F-pxna-EGFP* plasmids by LR clonase (11791020; Thermo Fisher Scientific).

Transgenic embryos were all generated by single-cell stage injection with each plasmid (250 ng/μl) and Tol2 transposase RNA (50 ng/μl; Kawakami et al., 2000).

### Generation of ZMEL cell line
ZMEL cell isolation was performed as previously described (Heilmann et al., 2015). Briefly, tumors were isolated from melanoma-bearing MiniCoopR fish and manually dissected in a dissection medium (50% F12, 50% DMEM, 10× pen/strep, 0.075 mg/ml Liberase) for 30 min at room temperature. After adding inactivating solution (50% F12, 50% DMEM, 10× pen/strep, 15% FBS) to the tumor suspension, tumor cells were filtered through a 40-μm strainer (08-771-1; Thermo Fisher Scientific) three times and then centrifuged for 5 min at 500 rcf and resuspended with 2 ml of complete media (see details in Heilmann et al., 2015) and plated on fibronectin-coated wells of a 6-well plate. Cells were monitored for 2 wks. Once they adhered and started to proliferate, cells were passaged to a 10-cm plate and flow cytometry was performed to sort for GFP + cells. Primary ZMEL cells were cultured in complete media with 5% CO$_2$ at 28.5°C. After ~10 passages, cells began to proliferate readily and ZMEL cells could then be cultured in standard ZMEL media (DMEM with 10% FBS and 1× glutaMAX).

### ZMEL cell transplantation into zebrafish larvae
ZMEL cell transplantations were performed as previously described (Roh-Johnson et al., 2017). Briefly, ZMEL cells were harvested and resuspended in HBSS at 10$^6$ cells/ml. Cells were loaded into a microinjection needle and 50 nl of the ZMEL cell suspension was transplanted into the hindbrain ventricle of anesthetized 2 dpf zebrafish larvae by using an oil-controlled microinjection rig (4r Oil, #5196000030; Eppendorf CellTram) with a Narishige arm. Injected larvae were incubated at 28.5°C. Imaging was performed 1–4 d after transplantation using a Leica Yokogawa CSU-W1 spinning disc confocal microscope in a 28.5°C environmental chamber.

### ZMEL transfection
To exogenously express Paxillin Y118 mutants in ZMEL cells, 6 million ZMEL-mCherry cells were transfected with 30 μg *mEGFP-WT*-zebrafish *pxna*, *mEGFP-Y118E*-zebrafish *pxna*, or *mEGFP- Y118F*-zebrafish *pxna* plasmid, respectively, using Neon Transfection System (catalog #MPK10096; Thermo Fisher Scientific) under 1,200 V, 20 ms, and 2 pulses conditions. Transfected cells were then FACS-isolated for similar expression levels of GFP (~10-fold overexpression over endogenous Paxillin).

### Zebrafish live imaging
Larval zebrafish were maintained at 28.5°C in E3 medium with 0.003% N-Phenylthiourea (P9629; Sigma-Aldrich) to prevent pigmentation. For live imaging assays, 3–6 dpf larval zebrafish were anesthetized using 0.2 mg/ml Tricaine-S (NC0872873; Thermo Fisher Scientific) solution, and mounted in 1% low melting agarose (#16520-050; Thermo Fisher Scientific) in a 35-mm glass bottom dish (FD35-100, World Precision Instruments) for imaging. Imaging was performed using either a PL APO 40×/1.10 water immersion objective or a PL APO 63×/1.40 oil immersion objective with 1× or 2× zoom on a Leica Yokogawa CSU-W1 spinning disc confocal microscope with iXon Life 888 EMCCD camera at 28.5°C.

### ZMEL cell migration velocity analysis
ZMEL cells in a culture undergoing random cell migration were imaged every 5 min for 10 h. ZMEL cell migration in vitro and in vivo was tracked with the Manual Tracking plugin on FIJI software under maximum intensity Z-stack projection. Final velocities were calculated by averaging the velocities from each time point of a time-lapse recording. Mean squared displacements were quantified using the flowcatchR package (Marini and Binder, 2018). Directionality ratios were defined as d/D, where d is the straight-line migration distance between the start and the end timepoints and D is the total distance that cells migrate within a given time. D and d values are calculated based on the x-y coordinates of cell migration trajectory from the Manual Tracking plugin on FIJI software.

### Macrophage-directed migration analysis
To perform directed migration assay, *Tg(mpeg1:WT-Paxillin-EGFP)*, *Tg(mpeg1:Y118E-Paxillin-EGFP)*, or *Tg(mpeg1:Y118F-Paxillin-EGFP)* zebrafish lines were crossed with *Tg(mfap4:tdTomato-CAAX)^xt6*. Tail

wound transections were performed on 3 dpf larvae with a size 10 scalpel as previously described (Barros-Becker et al., 2017). Macrophage recruitment to the wound was imaged within 4 h of wounding with standard zebrafish live imaging approaches (see above).

To analyze directed migration velocity, zebrafish macrophages were selected by thresholding the fluorescence intensity under maximum intensity Z-stacks projections using the RFP channel on FIJI software. The x-y coordinates of the cell centroid were tracked over time to calculate migration velocities. The final velocity was calculated by averaging the velocities from each time point of a time-lapse recording.

To plot cell migration trajectories, the x and y coordinates of macrophage centroids from the above velocity analyses were used to track macrophage migration with R (version 3.6.1) and RStudio (version 1.4.1106). Macrophage locations at timepoint 0 were normalized to the same initial coordinates, (x = 0, y = 0). Trajectories from fish of the same genotype were plotted in the same graph. The x and y coordinates were adjusted in some fish to consistently illustrate directed migration towards the x axis. The code for macrophage migration analysis is available on GitHub (https://github.com/rohjohnson-lab/Xue_2022). Mean squared displacements were quantified using the flowcatchR package (Marini and Binder, 2018).

### Focal adhesion size analysis
Focal adhesion sizes were measured by thresholding the Paxillin positive puncta with fluorescence intensity under maximum intensity Z-stacks projections using the GFP channel on FIJI software.

### Focal adhesion molecular dynamic analysis
FRAP was performed on a Leica Yokogawa CSU-W1 spinning disc confocal microscope equipped with a 2D-VisiFRAP Galvo System Multi-Point FRAP/Photoactivation module. ZMEL Paxillin-EGFP cells or transplanted larvae were both imaged at 28.5°C using a Okolab stage top incubator. Single z-plane images were taken every 2 s. Three frames were taken prebleach and 5 min of imaging were acquired postbleach. A region of interest (ROI) was drawn around individual Paxillin-positive punctae of ∼2 µm × 2 µm for photobleaching. The FRAP laser configuration was as follows: 405 nm laser line; 20 mW; 5 ms for 1 cycle.

FRAP analysis is performed as previously described (Legerstee et al., 2019). Briefly, using FIJI software, an ROI was drawn closely around the bleached area of individual Paxillin-positive punctae, as well as an unbleached non-cell area for background control. The mean fluorescence intensity for both bleached regions and control regions was measured. Relative fluorescence intensity (RFI) was background-subtracted and normalized to prebleached levels using RFI = $(I_t - I_{bgt}) / (I_{pre} - I_{bgpre})$, where $I_t$ is the mean fluorescence intensity of bleached area at time point t and $I_{bgt}$ is the mean fluorescence intensity of unbleached background area at time point t. $I_{pre}$ is the mean fluorescence intensity of the Paxillin-positive area at the prebleach timepoints and $I_{bgpre}$ is the mean fluorescence intensity of the background area at prebleach timepoints.

### Focal adhesion lifetime analysis
Focal adhesion live imaging was performed on a Leica Yokogawa CSU-W1 spinning disc confocal microscope for lifetime measurements. ZMEL Paxillin-EGFP cells or transplanted larvae were both imaged at 28.5°C using a Okolab stage top incubator. Images were taken every 20 or 30 s for 1 h, with 0.22 µm z-series for an approximate 5 µm z-depth.

To analyze focal adhesion lifetime, using FIJI software, z-slices comprising 2 µm of the cell ventral surface were merged as maximum intensity projections, then an ROI was drawn closely around a focal adhesion punctum, as well as a non-cell area for background control. The mean fluorescence intensity $(I_t)$ of an ROI was measured at all time points from a punctum beginning to assembly until full disassembly, and the background signal was subsequently subtracted. $I_t$ was further processed with a three-frame running average, and a lifetime curve was generated on Excel. Curve fitting was performed as previously described (Stehbens and Wittmann, 2014). Briefly, using "Solver" add-in on Excel, the focal adhesion assembly curve was fit into logistic function and the disassembly curve was fit into single exponential decay. Assembly and disassembly rates were calculated by the steepness of the curves and lifetime was further calculated based on the t-half of the assembly and disassembly fits.

### Immunostaining zebrafish larvae
5–7 dpf zebrafish embryos were fixed overnight at 4°C in 4% PFA. The next day, embryos were washed three times in PBS/0.1% Tween 5 min, and then permeabilized with 0.1% proteinase K (EO0491; Thermo Fisher Scientific) in PBS for 15 min at room temperature. Embryos were then fixed in 4% PFA for 20 min and washed five times with PBDT (1% BSA, 1% DMSO, and 0.5% Triton X-100 in PBS) for 5 min each. The embryos were blocked in PBDT with 10% goat serum for 2 h before incubating in a primary antibody at 4°C overnight. The embryos were then washed six times with PBDT for 15 min at room temperature and then incubated in a secondary antibody overnight at 4°C. Embryos were dehydrated step-wise in a 25, 50, and 75% glycerol series and were dissected and mounted for imaging. Primary antibodies used were chicken anti-GFP (1:500, ab13970; Abcam), mouse anti-mCherry (MCA-1C51; Encor Biotechnology), rabbit anti-pY118 Paxillin (1:500; NBP2-24459; Novus Biologicals), rabbit anti-Laminin (1:500, L9393; Sigma-Aldrich), and rabbit anti-Fibronectin (1:400; , F3648; Sigma-Aldrich). Secondary antibodies used were Alexa Fluor 488 goat anti chicken IgY(H + L) secondary antibody (1:500, 103-545-155; Jackson Immunoresearch), Alexa Fluor 633 goat anti rabbit IgG(H + L) secondary antibody (1:500, A-21070; Thermo Fisher Scientific), and DAPI (D9542; Sigma-Aldrich). Imaging was performed using a 63×/1.40 oil immersion objective on a Zeiss LSM 880 with three photomultiplier tubes and two GaSP detectors at room temperature (Carl Zeiss).

### Immunostaining cells in culture
1 d before immunostaining, ZMEL-GFP cells and ZMEL-mCherry cells that express WT-Paxillin were plated on either glass bottom dishes (FD35-100; World Precision Instruments) or collagen-

coated dishes (0.5 kPa, SV3520-COL-0.5; 50 kPa, SV3520-COL-50; Matrigen). Cells were fixed with 4% PFA and permeabilized in 0.1% Triton X-100/TBS for 5 min and then blocked with 1% BSA + 1% FBS for 1 h. Cells were incubated with primary antibodies overnight at 4°C and secondary antibodies for 1 h at room temperature. Primary antibodies used were chicken anti-GFP (1:500, ab13970; Abcam) and rabbit anti-pY118 Paxillin (1:500, NBP2-24459; Novus Biologicals). Secondary antibodies used were Alexa Fluor 488 goat anti-chicken IgY(H + L) secondary antibody (1:500, 103-545-155; Jackson Immunoresearch), Alexa Fluor 633 goat anti-rabbit IgG(H + L) secondary antibody (1:500, A-21070; Thermo Fisher Scientific), and DAPI (D9542; Sigma-Aldrich). Imaging was performed using either a Zeiss LSM 880 (Carl Zeiss) with a 63×/1.40 oil immersion objective with three photomultiplier tubes and two GaSP detectors or a Leica Yokogawa CSU-W1 spinning disc confocal microscope with a Leica PL APO 63×/1.40 oil immersion objective and an iXon Life 888 EMCCD camera at room temperature.

### YUMM1.7 cell culture and transduction
YUMM1.7 cells were cultured in standard media (DMEM with 10% FBS and non-essential amino acids) in 5% $CO_2$ at 37°C. To generate stably expressing YUMM1.7 cells, lentiviruses were produced by cotransfecting 293FT cells (R70007; Thermo Fisher Scientific) with mammalian Paxillin *p.LKO-WT/Y118E/Y118F-Paxillin-T2A-GFP* or mammalian FAK *p.LKO-GFP-FAK* and GFP control plasmid *p.LKO-GFP*, together with psPAX2 and VSV-G plasmids. Viral supernatant was collected 36 h after transfection and filtered through a 0.4-μm syringe filter. 50,000 parental YUMM1.7 cells were then transduced with 0.5 ml viral supernatant plus 10 μg/ml polybrene (TR-1003-G; Sigma-Aldrich) on a well of a 6-well dish for 48 h. Finally, cells were FACS-isolated for similar expression levels of GFP.

### Mouse tumor generation
YUMM1.7 cells stably expressing mammalian *WT-Paxillin-T2A-GFP*, *Y118E-Paxillin-T2A-GFP*, and *Y118F-Paxillin-T2A-GFP* were harvested, and 250,000 cells were injected subcutaneously into 6–8 wk old female C57BL/6J mice (000664; Jackson Laboratories). Once tumors reached ∼1 cm³, primary tumors were resected and collected for protein analysis.

### Western blot analysis
Cells in culture were lysed in RIPA lysis buffer (89900; Thermo Fisher Scientific) with proteinase and phosphatase inhibitor cocktail (P8340, 524635; Sigma-Aldrich) directly from cell culture plates. Tumor tissue was dissociated by OMNI tissue homogenizer and then lysed in the same conditions as described above. Lysates were centrifuged at 14,000 *g* in 4°C and the protein concentration was measured with Pierce BCA Protein Assay Kit (23225; Thermo Fisher Scientific).

15 μg protein was mixed with 4× Laemmli sample buffer (1610747; Bio-Rad) with a reducing agent and was separated by sodium dodecyl sulfate-polyacrylamide gel electrophoresis (SDS-PAGE). A 45 μm nitrocellulose membrane (1620115; Bio-Rad) was used to transfer protein from SDS-polyacrylamide gels. The blocking buffer was either 5% BSA (in TBS/0.1% TWEEN 20)

for detecting phospho-specific protein or 5% non-fat dry milk (in TBS/0.1% TWEEN 20) for non-phospho-specific protein. Primary antibodies used were rabbit anti-Paxillin (1:1,000, STJ94969; Antibodyplus), rabbit anti-pY118-Paxillin (1:1,000, 9369; Cell Signaling Technology), rabbit anti-FAK (1:1,000, 3285; Cell Signaling Technology), rabbit anti-pFAK397 (1:1,000, 3283; Cell Signaling Technology), chicken anti-GFP (1:500, ab13970; Abcam), mouse anti-CrkII (1:1,000610035; BD Bioscience), mouse anti-DOCK180 (1:500, sc-13163; Santa Cruz Biotechnology), mouse anti-C3G (1:250, sc-178403; Santa Cruz Biotechnology), rabbit p-ERK (1:1,000, 9101S; Cell Signaling Technology), and rabbit anti-pY31-Paxillin (1:1,000, 44-720G; Thermo Fisher Scientific). Secondary antibodies used were goat anti-chicken IgY(H + L) conjugated with HRP (1:10,000, A16054; Thermo Fisher Scientific), goat anti-mouse IgG(H + L) antibody conjugated with HRP (1:10,000, A28177; Thermo Fisher Scientific), and donkey anti-rabbit IgG(H + L) antibody conjugated with HRP (1:10,000, NA934V; GE Healthcare). Antibody signals were detected using SuperSignal West Pico Plus Chemiluminescent Substrate (34580; Thermo Fisher Scientific) and were further quantified using densitometric analysis on FIJI software.

### Co-immunoprecipitation
1,000 μg total protein was incubated with 5 μg rabbit anti-Paxillin (STJ94969; antibodyplus, isotype IgG) overnight at 4°C. The next day, 1.5 mg precleared Dynabeads Protein G (10003D) were added to the protein-antibody mixture and rotated for 2 h at 4°C, followed by three washes with RIPA lysis buffer. Finally, the protein was eluted using 2× Laemmli sample buffer (1610747; Bio-rad) with a reducing agent. Samples were further analyzed by Western blot.

### Graphical representations and statistical analysis
All graphs were generated from Prism (v7, GraphPad), Excel (v16.43, Microsoft), R (v3.6.1, R Development Core Team 2020), and RStudio (v1.4.1106, RStudio Team 2020). Statistical analyses were performed using Prism (v7, GraphPad), R (v3.6.1, R Development Core Team 2020), and RStudio (v1.4.1106, RStudio Team 2020). We performed repeated measures ANOVA as statistical tests to take into account both the variability within the technical replicates and the biological replicates. Specific statistical tests are indicated in each figure legend.

At this time, we are not considering sex as a biological factor, as all of our mouse studies were performed in female mice, and all of our zebrafish experiments were performed in larvae that had not yet undergone sex determination.

### Online supplemental material
Fig. S1 shows ZMEL cells form Paxillin-positive focal adhesion structures in vitro and in vivo. Fig. S2 shows Y118-Paxillin exhibits distinct phosphorylation status in migrating cancer cells in vivo versus in vitro. Fig. S3 shows Y118-Paxillin mutants localize to focal adhesion structures in ZMEL cells in the in vitro cell culture conditions. Fig. S4 shows ZMEL cells expressing Y118F-Paxillin exhibit increased cell migration in vivo. Fig. S5 shows there is no change in C3G or ERK activity in cells expressing Y118F-Paxillin in vivo. Video 1 shows migrating ZMEL

cells in the zebrafish skin forms Paxillin-positive structures colocalizing with actin. Video 2 shows migrating ZMEL cells in vivo transduce force to the environmental ECM. Video 3 shows Paxillin FRAP in ZMEL cells in vitro. Video 4 shows Paxillin FRAP in ZMEL cells in vivo. Video 5 shows Paxillin lifetime in ZMEL cells in vitro. Video 6 shows Paxillin lifetime in ZMEL cells in vivo. Video 7 shows that macrophages expressing Y118F-Paxillin exhibit increased directed cell migration toward the wound in vivo.

## Acknowledgments

We thank all members of the Roh-Johnson lab for discussions; Rodney Stewart and Paul Martin for zebrafish reagents; members of the Matthew Miller lab for helpful discussions; the Centralized Zebrafish Animal Resources (CZAR) at the University of Utah for zebrafish husbandry and equipment; the University of Utah Cell Imaging Core for use of Leica Yokogawa CSU-W1 spinning disc confocal microscope; the University of Utah Flow Cytometry Core for cell sorting; Huntsman Cancer Institute Preclinical Research Resource for mousee work; and the University of Utah Electron Microscopy Core for help with transmission electron microscopy.

This work was funded by National Institutes of Health grants R00CA190836 (to M. Roh-Johnson) and 1G20OD018369-01 (to Centralized Zebrafish Animal Resource).

Author contributions: Conceptualization, Q. Xue and M. Roh-Johnson; Methodology, Q. Xue and M. Roh-Johnson; Investigation, Q. Xue, S.R.S. Varady, T.Q. Alaka'i Waddell, M.R. Roman, J. Carrington, and M. Roh-Johnson; Formal Analysis, Q. Xue, S.R.S. Varady, and M.R. Roman; Writing, Q. Xue and M. Roh-Johnson; Funding acquisition, M. Roh-Johnson.

Disclosures: The authors declare no competing interests exist.

Submitted: 15 June 2022

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

## Supplemental material

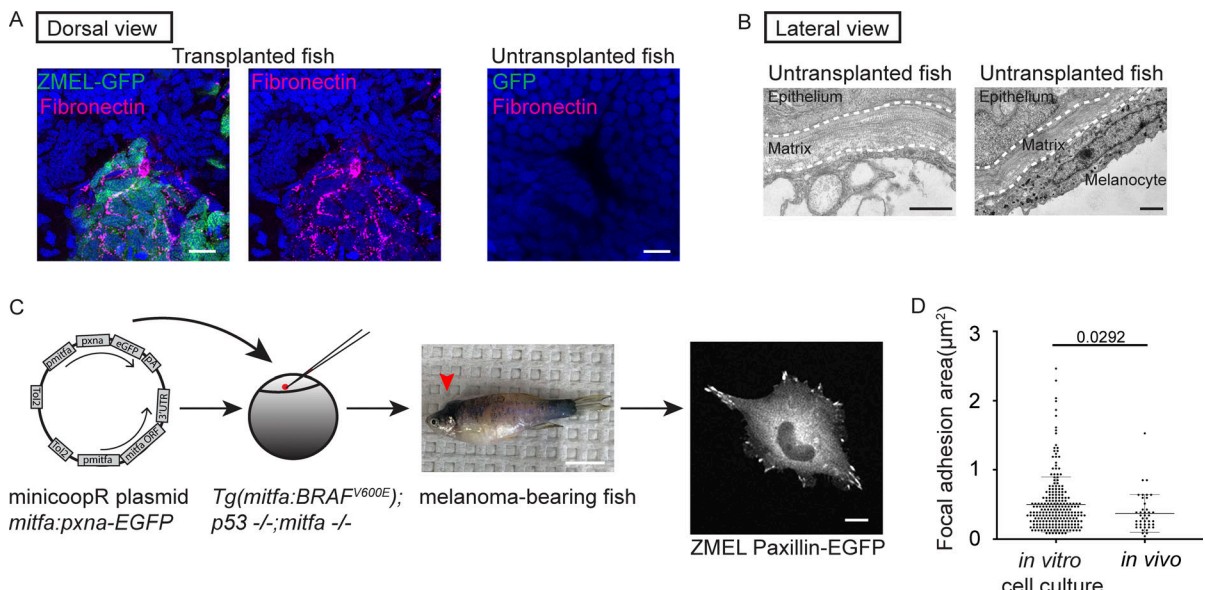

Figure S1.  **ZMEL cells form Paxillin-positive focal adhesion structures in vitro and in vivo. (A)** Representative examples of positive fibronectin im-munostaining of zebrafish larvae with transplanted ZMEL-GFP cells (left). Untransplanted larvae are similarly immunostained and used as a control to show the absence of fibronectin immunostaining in the same tissue (right). Scale bar is 10 μm. **(B)** Two TEM micrographs of untransplanted larvae at the skin region, 5 dpf, lateral view. Dashed white lines outline the skin ECM. Left panel indicates the absence of cells underneath the matrix and right panel indicates a pigmented melanocyte underneath the matrix. Scale bars are 1 μm. **(C)** Schematic of the process to generate primary ZMEL Paxillin-EGFP lines. A MiniCoopR plasmid expressing GFP-tagged zebrafish paxillin-a (pxna) is injected into single-cell stage embryos of Tg(mitfa:BRAF V600E); p53(lf); mitfa(lf). The melanocyte-rescued larvae are sorted and raised into adulthood for melanoma development (red arrowhead indicates melanoma tumor on adult zebrafish, middle panel). ZMEL Paxillin-EGFP cells are isolated from zebrafish melanoma tumors and cultured in cell culture dishes in vitro. Live imaging reveals Paxillin localizes to focal adhesions under in vitro cell culture conditions. Scale bar is 1 cm for melanoma-bearing zebrafish and 10 μm for the ZMEL Paxillin-EGFP cell. **(D)** Quantification of focal adhesion size in ZMEL Paxillin-EGFP cells in culture compared with in vivo. Non-parametric unpaired $t$ test, Mean ± SD. $n$ = 234 focal adhesions (12 cells) in vitro and $n$ = 42 focal adhesions in vivo (4 cells).

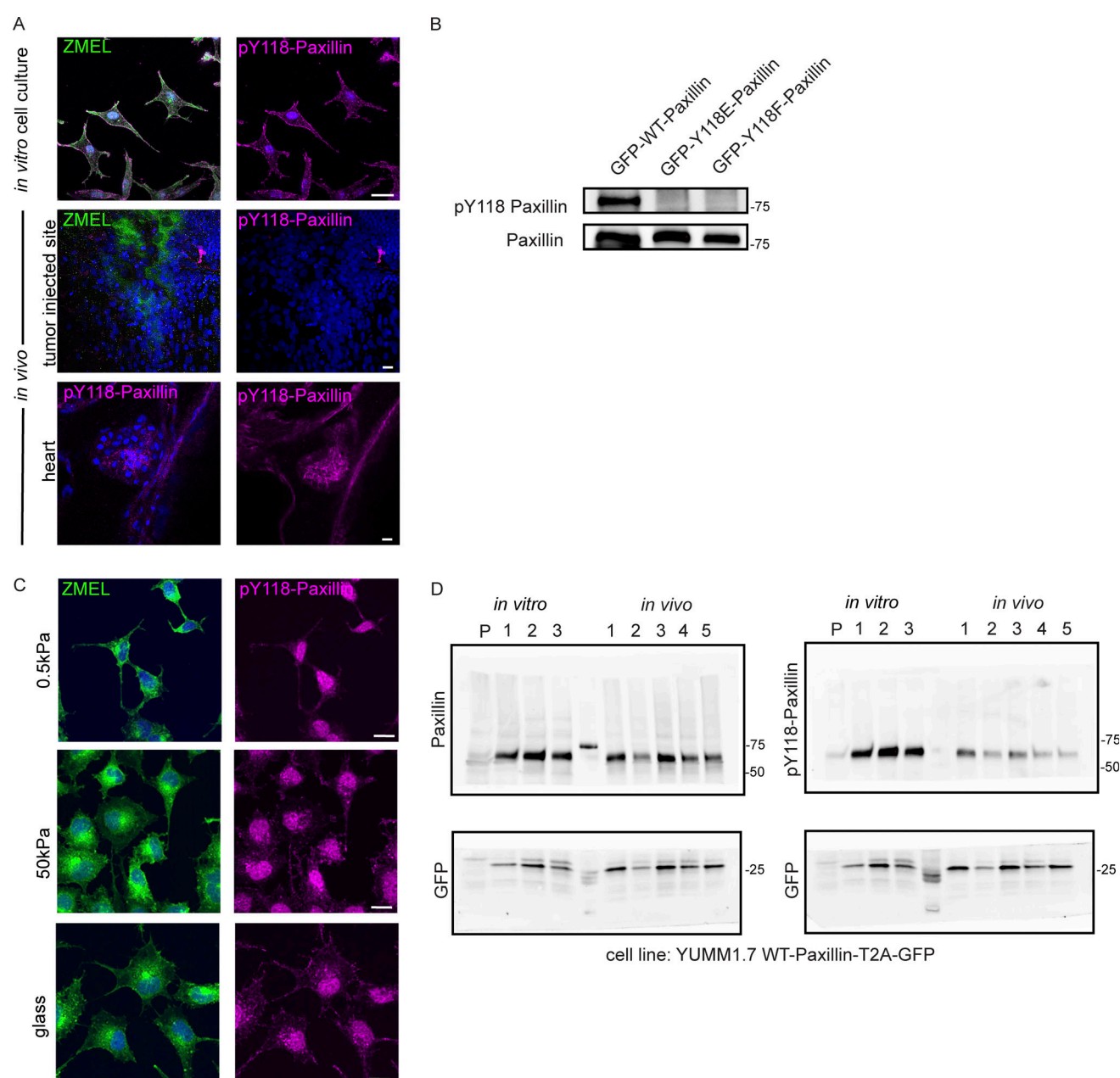

Figure S2. **Y118-Paxillin exhibits distinct phosphorylation status in migrating cancer cells in vivo versus in vitro. (A)** Top: pY118-Paxillin immunostaining (magenta) of ZMEL-GFP (GFP immunostaining, green) plated on in vitro cell culture dishes. Middle: pY118-Paxillin immunostaining (magenta) of ZMEL-mCherry (mCherry immunostaining, pseudo-colored green) in larval zebrafish (3 d post-transplantation). Bottom: pY118-Paxillin immunostaining (magenta) of the zebrafish developing heart (5 dpf). **(B)** Western blot showing the specificity of the pY118-Paxillin antibody and that it does not recognize Y118E-Paxillin and Y118F-Paxillin. **(C)** Representative images of ZMEL-GFP cells plated on 2D surfaces of different stiffnesses (left) and stained for pY118-Paxillin (right). **(D)** Unmodified Western blot of panels shown in Fig. 3 C—YUMM1.7 cells plated in culture and YUMM1.7 melanoma tumors in vivo blotted with Paxillin and pY118-Paxillin antibodies. "P" is parental cell line with no GFP expression. GFP was used as the loading control and as a control for the number of YUMM1.7 cells in mouse tumors. Source data are available for this figure: SourceData FS2.

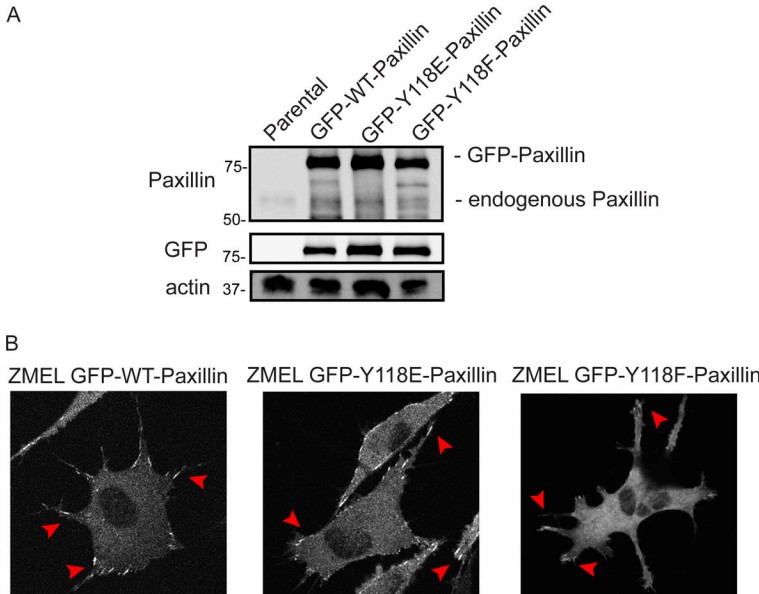

Figure S3.   **Y118-Paxillin mutants localize to focal adhesion structures in ZMEL cells in the in vitro cell culture conditions. (A)** Western blot revealing relative expression levels of endogenous Paxillin (top panel) and GFP-tagged Paxillin (top panel and middle GFP panel) in ZMEL cells overexpressing mutant and wildtype variants of GFP-tagged Paxillin. Actin is used as a loading control. **(B)** Representative live images of ZMEL cells expressing GFP-WT/Y118E/Y118F-Paxillin in the in vitro cell culture conditions. Red arrowheads indicate Paxillin-positive focal adhesion structures. Scale bar is 10 µm. Source data are available for this figure: SourceData FS3.

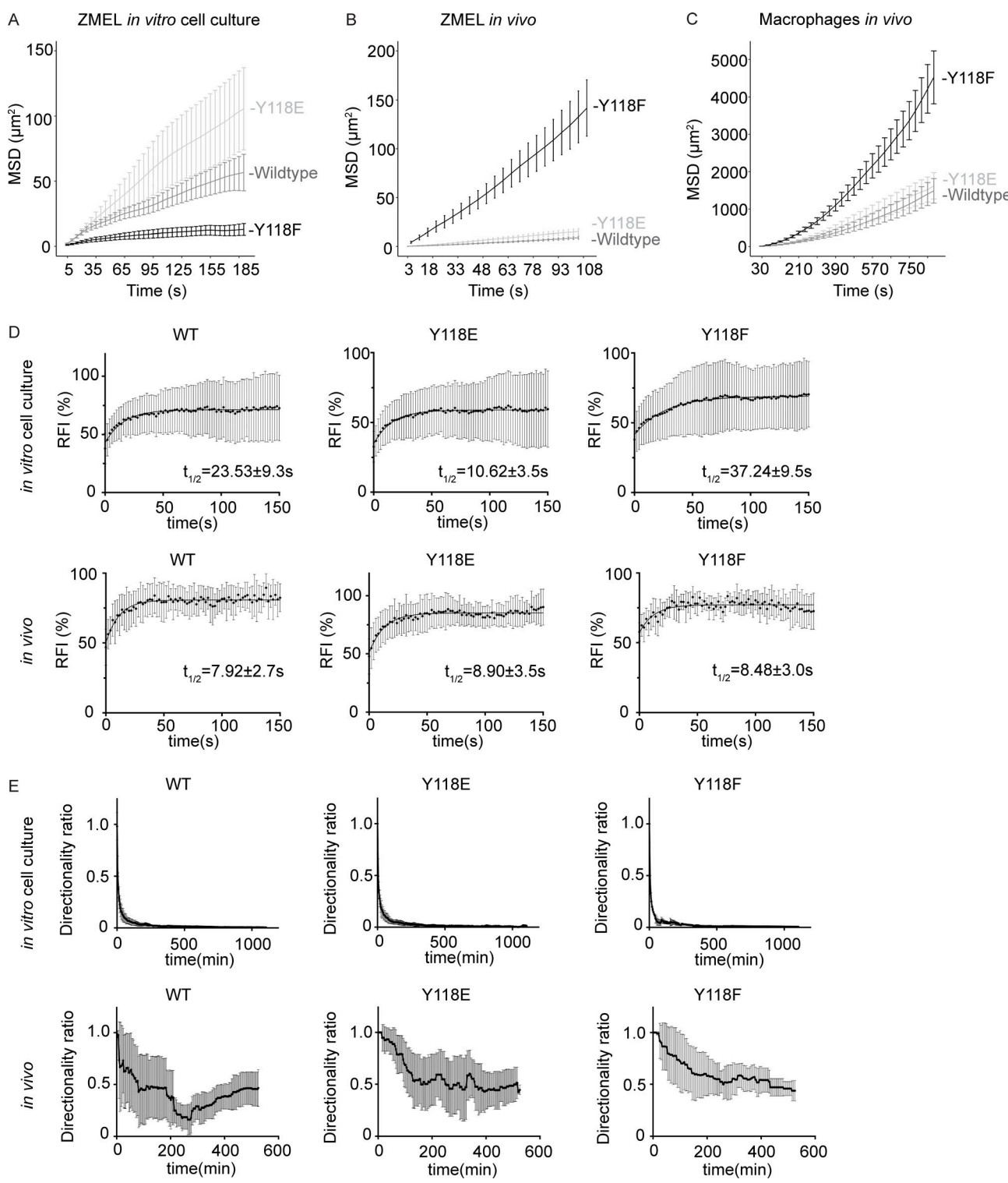

Figure S4. **ZMEL cells expressing Y118F-Paxillin exhibit increased cell migration in vivo. (A–C)** Mean squared displacement (MSD) measurements for ZMEL cells expressing WT, Y118E, or Y118F-Paxillin in cell culture (A); ZMEL cells expressing WT, Y118E, or Y118F-Paxillin in vivo (B); and zebrafish macrophages expressing WT, Y118E, or Y118F-Paxillin in vivo (C). **(D)** Cumulative FRAP recovery curves of ZMEL cells expressing WT, Y118E, or Y118F-Paxillin in the in vitro cell culture conditions and ZMEL cells expressing WT, Y118E, or Y118F-Paxillin in vivo after photobleaching. $n$ = 34, 44, and 51 cells for WT, Y118E, Y118F in vitro, and $n$ = 7 cells/6 fish, 6 cells/6 fish, and 6 cells/5 fish for WT, Y118E, Y118F in vivo. Mean ± SD. **(E)** Directionality ratios of ZMEL cells expressing WT, Y118E, or Y118F-Paxillin in the in vitro cell culture conditions and ZMEL cells expressing WT, Y118E, or Y118F-Paxillin in vivo. $n$ = 32, 48, and 14 cells for WT, Y118E, Y118F in vitro, and $n$ = 8 cells/3 fish, 12 cells/3 fish, and 15 cells/3 fish for WT, Y118E, Y118F in vivo. Mean ± SD.

Figure S5. **FAK is downregulated in vivo compared to in vitro, and there is no change in C3G or ERK recruitment to Paxillin in cells expressing Y118F-Paxillin in vivo. (A)** Unmodified Western blot for Fig. 5 B—YUMM1.7 cells plated in culture and YUMM1.7 melanoma tumors (in vivo) blotted with FAK and pY397-FAK antibodies. GFP was used as the loading control, and as a control for the number of YUMM1.7 cells in mouse tumors. **(B)** YUMM1.7 cells over-expressing GFP-FAK and Western blotted for FAK levels to confirm overexpression in vitro and in vivo. Actin is used as a loading control. **(C)** Co-immunoprecipitation analysis of Vinculin and Paxillin, immunoprecipitating Paxillin from YUMM1.7 tumors expressing wildtype, Y118E, and Y118F Paxillin and assaying for Vinculin interactions. **(D–I)** Co-immunoprecipitation analyses of C3G or activated ERK (p-ERK) with Paxillin in YUMM1.7 cell lines that ex-ogenously express mammalian wildtype, Y118E, and Y118F Paxillin in vitro (D–F) and in in vivo tumors (G–I). **(E and H)** Quantification of C3G/Paxillin ratio from D and G, bands from cells expressing wildtype Paxillin are normalized to 1 both in vitro and in vivo. $n = 3$ technical replicates. Non-parametric one-way ANOVA, error bars are mean ± SD. **(F and I)** Quantification of p-ERK/Paxillin levels from D and G, bands from cells expressing wildtype Paxillin are normalized to 1 both in vitro and in vivo. $n = 3$ technical replicates. Non-parametric one-way ANOVA, error bars are mean ± SD. **(J)** Western blot analysis of pY31-Paxillin in mouse melanoma YUMM1.7 cells expressing mammalian WT-Paxillin-T2A-GFP plated on the in vitro cell culture dishes ($n = 3$ dishes) and YUMM1.7 melanoma in vivo tumors ($n = 5$ tumors). GFP was used as the loading control and a control for number of YUMM1.7 cells in mouse tumors. **(K)** Quantification of pY31-Paxillin/total Paxillin protein ratio from J. Error bars are mean ± SD. Non-parametric unpaired $t$ test. Source data are available for this figure: SourceData FS5.

Video 1. **Migrating ZMEL cells in the zebrafish skin form Paxillin-positive structures colocalizing with actin.** Related to Fig. 1. Timelapse video of a ZMEL cell co-expressing Paxillin (Paxillin-EGFP, green) and actin (Lifeact-mScarlet, magenta) migrating in the larval zebrafish skin, maximum intensity pro-jection of Z-stacks. Images were taken every 30 s for 30 min, 15 fps.

Video 2.   **Migrating ZMEL cells in vivo transduce force to the environmental ECM.** Related to Fig. 1. Timelapse video of ZMEL-mCherry cells (magenta) migrating in the skin of a zebrafish larva expressing GFP-labelled collagen *Tg(krt19:col1a2-GFP)*<sup>zj502</sup>, maximum intensity projection of Z-stacks. Arrow indicates the bending event of a collagen fiber toward the migrating cell, suggesting that the migrating cell is pulling on the collagen fiber. Images were taken every 2 min for 3 h, 20 fps.

Video 3.   **Paxillin FRAP in ZMEL cells in vitro.** Related to Fig. 2. Example of a FRAP video of a ZMEL Paxillin-EGFP cell in the in vitro cell culture conditions. A single Paxillin-positive punctum (as indicated by the red circle) is bleached, and fluorescence recovery after bleaching is recorded as a single Z-plane. Images were taken every 2 s for 5 min, 7 fps.

Video 4.   **Paxillin FRAP in ZMEL cells in vivo.** Related to Fig. 2. Example of a FRAP video of a ZMEL Paxillin-EGFP cell that is transplanted in vivo. A single Paxillin-positive punctum (as indicated by the red circle) is bleached, and fluorescence recovery after bleaching is recorded as a single Z-plane. Images were taken every 2 s for 5 min, 7 fps.

Video 5.   **Paxillin lifetime in ZMEL cells in vitro.** Related to Fig. 2. Timelapse video of a single Paxillin-positive punctum from a ZMEL Paxillin-EGFP cell that is plated in the in vitro cell culture conditions, maximum intensity projection Z-stacks of cell ventral surface. Images were taken every 30 s for 1 h, 7 fps.

Video 6.   **Paxillin lifetime in ZMEL cells in vivo.** Related to Fig. 2. Timelapse video of a single Paxillin-positive punctum from a ZMEL Paxillin-EGFP cell that is transplanted in vivo, maximum intensity projection Z-stacks of cell ventral surface. Images were taken every 20 s for 1 h, 7 fps.

Video 7.   **Macrophages expressing non-phosphorylatable (Y118F) Paxillin exhibit increased directed cell migration toward the wound in vivo.** Related to Fig. 4. Side-by-side timelapse videos of macrophage-directed cell migration in (left) *Tg(mpeg:Paxillin-wt-EGFP)*<sup>zj503</sup>; (middle) *Tg(mpeg:Paxillin-Y118E-EGFP)*<sup>zj504</sup>; and (right) *Tg(mpeg:Paxillin-Y118F-EGFP)*<sup>zj505</sup> zebrafish larvae, maximum intensity projection of Z-stacks. Macrophages were tracked by Manual Tracking plugin on FIJI. Images were taken every 30 s for 10 min, 7 fps.

