## [Peer Review File · The Journal of Cell Biology]

Lack of Paxillin phosphorylation promotes single cell migration in vivo

Qian Xue, Sophia Varady, Trinity Waddell, Mackenzie Roman, James Carrington, and Minna Roh-Johnson

Corresponding Author(s): Minna Roh-Johnson, University of Utah and Trinity Waddell, University of Utah

Review Timeline:

Submission Date:	2022-06-15
Editorial Decision:	2022-07-22
Revision Received:	2022-11-30
Editorial Decision:	2022-12-28
Revision Received:	2023-01-05

Monitoring Editor: Anna Huttenlocher

Scientific Editor: Lucia Morgado-Palacin

Transaction Report:

DOI: <https://doi.org/10.1083/jcb.202206078>

July 22, 2022

Re: JCB manuscript #202206078

Minna Roh-Johnson
University of Utah
Biochemistry
15N Medical Drive East
Salt Lake City, UT 84112-5650

Dear Dr. Roh-Johnson,

Thank you for submitting your manuscript entitled "Lack of Paxillin phosphorylation promotes single cell migration in vivo". The manuscript was assessed by expert reviewers, whose comments are appended to this letter. We invite you to submit a revision if you can address the reviewers' key concerns, as outlined here.

As you will see, the reviewers are overall enthusiastic about this in vivo study, and we, the editors, feel the same. However, the reviewers raise some overlapping concerns that would need to be further addressed with new experimentation, which we agree with.

Reviewers #2 and #3 feel the functional relationship between the migration velocities, the dynamics of focal adhesions and the status of Paxillin(Y118) phosphorylation needs additional investigation, and thus they request that you examine the turnover/assembly/disassembly of focal adhesions in the phosphorylation mutants in vivo by FRAP assays. We acknowledge that quantification of dynamics is challenging in vivo, but these analyses would raise the impact of the paper. It is also particularly important, as noted by reviewer #2, that you address the level of expression of the constructs in vivo and report if you see a difference with the bright and dim cells.

In addition, reviewer #2 thinks more evidence is required to support the proposed mechanism in YUMM1.7 cells that lack of Paxillin(Y118) phosphorylation activates the CRKII-DOCK180-Rac pathway, which we agree with -for instance, the requirement of Paxillin(Y118) phosphorylation for CRKII binding and the activation of Rac signaling needs better demonstration.

Reviewers #1 and #3 also ask whether the described phenotypes are tissue-specific (would like that you study focal adhesions in a second tissue), and about the nature of the substrate that the cells are migrating in the living animal. Our view is that including a second tissue is not strictly required for resubmission, but the study of focal adhesions in zebrafish macrophages, for example, which you showed that exhibit enhanced migration in the absence of Paxillin(Y118) phosphorylation, is highly recommended for a successful revision.

We hope that you will be able to address each of the reviewers' other points, though.

GENERAL GUIDELINES:

Text limits: Character count for an Article is < 40,000, not including spaces. Count includes title page, abstract, introduction, results, discussion, and acknowledgments. Count does not include materials and methods, figure legends, references, tables, or supplemental legends.

Figures: Articles may have up to 10 main text figures. Figures must be prepared according to the policies outlined in our Instructions to Authors, under Data Presentation, <https://jcb.rupress.org/site/misc/ifora.xhtml>. All figures in accepted manuscripts will be screened prior to publication.

Supplemental information: There are strict limits on the allowable amount of supplemental data. Articles may have up to 5 supplemental figures. Up to 10 supplemental videos or flash animations are allowed. A summary of all supplemental material should appear at the end of the Materials and methods section.

Please note that JCB now requires authors to submit Source Data used to generate figures containing gels and Western blots

with all revised manuscripts. This Source Data consists of fully uncropped and unprocessed images for each gel/blot displayed in the main and supplemental figures. Since your paper includes cropped gel and/or blot images, please be sure to provide one Source Data file for each figure that contains gels and/or blots along with your revised manuscript files. File names for Source Data figures should be alphanumeric without any spaces or special characters (i.e., SourceDataF#, where F# refers to the associated main figure number or SourceDataFS# for those associated with Supplementary figures). The lanes of the gels/blots should be labeled as they are in the associated figure, the place where cropping was applied should be marked (with a box), and molecular weight/size standards should be labeled wherever possible. Source Data files will be made available to reviewers during evaluation of revised manuscripts and, if your paper is eventually published in JCB, the files will be directly linked to specific figures in the published article.

The typical timeframe for revisions is three to four months. While most universities and institutes have reopened labs and allowed researchers to begin working at nearly pre-pandemic levels, we at JCB realize that the lingering effects of the COVID-19 pandemic may still be impacting some aspects of your work, including the acquisition of equipment and reagents. Therefore, if you anticipate any difficulties in meeting this aforementioned revision time limit, please contact us and we can work with you to find an appropriate time frame for resubmission. Please note that papers are generally considered through only one revision cycle, so any revised manuscript will likely be either accepted or rejected.

Thank you for this interesting contribution to Journal of Cell Biology. You can contact us at the journal office with any questions, cellbio@rockefeller.edu.

Sincerely,

Anna Huttenlocher
Monitoring Editor
Journal of Cell Biology

Lucia Morgado-Palacin, PhD
Scientific Editor
Journal of Cell Biology

Reviewer #1 (Comments to the Authors (Required)):

Review of Xue et al.

Bulk of studies addressing focal adhesion dynamics were performed using culture models where cells are plated in 2D and in some cases 3D culture models. Several years, a controversy of the existence of focal adhesions in 3D was resolved but there really had been no clear-cut answer to the molecular mechanisms that govern focal adhesions in vivo. This is a beautiful study that finally addresses the mechanism of focal adhesion formation for endogenously tagged proteins in a living animal. Using a suite of transgenic zebrafish lines, high quality intravital imaging and clever molecular tools, the authors determine that there is a differential regulation of tyrosine 118 in the assembly of focal adhesions. In culture models, tyrosine 118 is phosphorylated during adhesion formation in migrating cells. Here, the authors determined that this site shows a reduction in phosphorylation activity in functional adhesions that are formed during in vivo migration. The study is well done and the data are convincing. I have really minor comments detailed below:

- 1) For the in vivo imaging, it would be interesting to understand the chemistry/composition of the substrate that these cells are migrating. Staining for ECM markers can address this.
- 2) It is intriguing that there is a differential phosphorylation of tyrosine 118, if cells are placed in a different organ environment where presumably the physi-chemical composition is different, would that finding be conserved. E.g. if you place in the tail/fin of the zebrafish would it show the same regulation. I understand that the macrophages show increased motility due to the reduced phosphorylation to the wounded area. Instead, I am asking, once the cells get to that environment is there a switch to a different focal adhesion mechanism or is it conserved for the in vivo environments tested.

Minor quibble,

Graphs of motility can be shown as , where the plots can then be used to assess types of motility, persistence etc.

Reviewer #2 (Comments to the Authors (Required)):

In this manuscript, Qian Xue and colleagues explore the regulation and function of the focal adhesion paxillin during in vitro and in vivo migration. The past literature has pointed to important differences in focal adhesions' functions between in vivo 3D, soft environment migration and in vitro 2D, rigid substrate migration. This is a nice tentative to elucidate the molecular differences that are responsible for the generally weak adhesions observed in vivo. The authors show that paxillin Tyr118 phosphorylation is strongly reduced in the in vivo situation, possibly because of a low level of FAK. Moreover, they show that in these in vivo conditions, the low phosphorylation levels facilitates migration, whereas it inhibits 2D in vitro migration. They then attempt to determine how non-phosphorylatable paxillin could contribute to migration and propose that it activates the CRKII-DOCK180-Rac pathway. There are some interesting observations , however I found the last part of the manuscript less convincing first because it challenges the common idea that paxillin binding to CRKII requires Tyr118 phosphorylation to interact with the CRKII SH2 domain and therefore must be documented by some very strong convincing evidence. This is unfortunately not the case in particular because of technical imprecisions, which need to be clarified.

General comments

1- What is the link between the difference in focal adhesion and paxillin dynamics observed in figure 2 and the phosphorylation status of paxillin? How do the phosphorylation mutant affect the dynamics of focal adhesions? Is the slower dynamics responsible for the slower migration? Although this hypothesis is tempting, the change in focal adhesion dynamics may also be a consequence of the slower migration observed.

2- The proposed mechanism exposed remains very hypothetical at this stage and I am not sure figure 6 should be shown as many points are not actually demonstrated.

- The authors propose that FAK low expression level is responsible for the low level of phosphorylated paxillin. However, this is not demonstrated. Can FAK overexpression rescue paxillin phosphorylation?

- They also indicate that the non phosphorylated paxillin better activates the Rac GEF DOCK180. Can they show that Rac is overactivated?

- The drawing of the cell indicate a similar number of focal adhesion in both in vivo and in vitro situation. This does not seem to be the case from the images shown, but as not been quantified.

- Do the authors have any indication of an increased recruitment of DOCK180 and/or CRKII at focal adhesions? Since the focal adhesions appear very small and limited in number in vivo compared to in vitro, an increase in Rac activity would required a very strong concentration of these proteins. Could this signaling take place somewhere else?

3- Paxillin is known to interact with multiple signaling molecules through both its phosphorylated tyrosines Y31 and Y118. Even if I understand that the authors do not want to study the functional role of Y31 in this paper, can they simply show if Y31 phosphorylation is also downregulated in vivo? Alternative, relevant partners of paxillin and CRKII such as p120Ras GAP or C3G are not considered or discussed. Since Rac overactivation is not shown, it is difficult to rule these out.

4- One major concern is the relative expression level of GFP-paxillin constructs compared to the endogenous proteins in the different models.

- Since the constructs and the transfection protocols are different in the different settings, it would be important to show the complete paxillin westernblot (showing both the endogenous and the GFP tagged proteins) in figures. Moreover, for the mutant forms to have a dominant effect, there need to be a high level of expression compared to the endogenous.

- Also, can the authors also be clarify where the GFP tag is? From the mat and met section, it seems that in some constructs it is located in C-ter (as expected) but may be in N-ter in others. A N-ter GFP tag is likely to affect the interactions with paxillin partners, such as CRKII.

Specific comments

1- Figure 2. When characterizing the dynamics of focal adhesions, the authors should first quantify the number and size of focal adhesions, which is an important parameter to take into account when looking at the effect of FAK and paxillin activity.

2- Figure 3 and 4a. The immunofluorescence images in panel b are difficult to see. Zooms of specific regions of interest should be shown.

3- There is very little or no Phospho Y118 staining visible but this may be due to the very small and sparse focal adhesions observed in vivo. To further confirm the lack of phosphorylation, the authors should show P-Y118 staining together with total paxillin (GFP or immunofluorescence) or another marker of focal adhesions.

4- Line 268. The sentence suggests that the non-phosphorylatable paxillin reduces cell velocity compared to the wild type, whereas the graph shows that the velocity is decreased only when compared to the phospho-mimetic mutant. The authors cannot conclude (line 274) that the "non-phosphorylatable Y118 paxillin inhibits cell migration"

5- Figure 5. I am confused as to why is the total paxillin immunoprecipitated and not only the mutants. It is important to show the endogenous and GFP tagged paxillin in all the figure panels. Or are the mutants untagged here? In this case, the authors should redo the experiment with tagged version and GFP IP.

6- Fig 5. Panel d. What is FAK level normalized to? If it is to GFP, this does not seem the most appropriate. Another blot showing actin for instance could be used.

7- Figure 5. Quantification based on only 2 technical repeats is insufficient and cannot be used for statistical test. Biological repeats and more repeats in general are required.

8- For all graphs, please indicate the exact p values instead of the stars.

9- For the westernblot quantifications, the data corresponding to each experimental values should be shown in the graph (not only the mean+ error bars : s.d. or SEM?)

10- I think that for clarity, the different pieces of a same westernblot (for instance in 5b) should be shown as a single panel with a single line indicating the cut region. As it is, it appears as if coming from distinct blots and therefore not quantifiable.

Reviewer #3 (Comments to the Authors (Required)):

1. A short summary of the paper, including description of the advance offered to the field. If you feel that prior literature undermines any aspect of this work, specific references are appreciated.

Really interesting study that highlights fundamental molecular differences between in vivo and in vitro mechanisms of cell migration, challenging the dogma that has been created by in vitro studies, showing clearly that in vivo studies are fundamental.

The authors developed a zebrafish transplantation system to visualize focal adhesion structures during single cell migration in vivo with high-resolution live cell imaging and compared focal adhesion dynamics to the traditional in vitro cell culture model. By FRAP and monitoring the fluorescence recovery over time the authors found that Paxillin exhibits a significantly faster molecular turnover rate in vivo as compared to cells in vitro. Also, the assembly and disassembly rates differed between the in vivo and in vitro environments (in vivo higher assembly and reduced disassembly).

Then the authors "sought to identify the molecular regulation that might explain these differences" and found that Paxillin, tyrosine 118 (Y118), exhibits reduced phosphorylation in migrating cells in vivo in both zebrafish and mouse melanoma models, contrary to the pivotal role for this phosphorylation event in cell culture studies.

Modulation of this residue by site-directed mutagenesis leads to opposite cell migration phenotypes in vivo versus in vitro in both migrating cancer cells and macrophages (reduced phosphorylation in vivo increases cell velocity but in vitro it reduces velocity of the cells). FAK is downregulated in cells in vivo, but the downstream effectors DOK180 and CRKII are upregulated. Cells expressing non-phosphorylatable Y118-Paxillin exhibit increased activation in CRKII-DOCK180-Rac pathway, leading to increased cell migration.

2. For each main point of the paper, please indicate if the data are strongly supportive. If not, indicate the evidence you feel is required. Please highlight those points you feel are most crucial for further consideration at JCB.

By FRAP and monitoring the fluorescence recovery over time the authors found that Paxillin exhibits a significantly faster molecular turnover rate in vivo as compared to cells in vitro. Also, the assembly and disassembly rates differed between the in vivo and in vitro environments (in vivo higher assembly and reduced disassembly). OK

Then the authors "sought to identify the molecular regulation that might explain these differences" and found that Paxillin, tyrosine 118 (Y118), exhibits reduced phosphorylation in migrating cells in vivo in both zebrafish and mouse melanoma models, contrary to the pivotal role for this phosphorylation event in cell culture studies. (Fig.3)

Modulation of this residue by site-directed mutagenesis leads to opposite cell migration phenotypes in vivo versus in vitro in both migrating cancer cells and macrophages (reduced phosphorylation in vivo increases cell velocity but in vitro it reduces velocity of the cells). OK

I think this is where I found data was missing (or maybe I missed) showing that the non-phosphorylatable residue Y118F indeed had a >> turnover/assembly rate (could see this in vitro), or the constitutively active tyrosine phosphorylation(Y118E) had lower turnover (in vivo?).

FAK is downregulated in cells in vivo, but the downstream effectors DOK180 and CRKII are upregulated. Cells expressing non-phosphorylatable Y118-Paxillin exhibit increased activation in CRKII-DOCK180-Rac pathway, leading to increased cell migration. OK

3. Lastly, indicate any additional issues you feel should be addressed (text changes, data presentation, etc.)

Comments:

As a non-expert in cell migration, I apologize if some points were not clear to me.

1- By FRAP and monitoring the fluorescence recovery over time the authors found that Paxillin exhibits a significantly faster molecular turnover rate in vivo as compared to cells in in vitro. Also, the assembly and disassembly rates differed between the in vivo and in vitro environments (in vivo higher assembly and reduced disassembly).

For me was not so clear how you have a faster molecular turnover rate but have lower disassembly rate - can you please clarify?

- 2- In figure 3b, it would be nice to show that the Ab does not recognize the Y118F as a control - since what I understood this Ab was raised against mammalian Paxillin (and in this figure it is the zebrafish Paxillin- or pxna-that is being overexpressed).
- 3- It is not clear throughout the manuscript if the authors are talking about the zebrafish paxillin, mouse or human. I had to go through the methods/legends..it would be useful to put zPaxillin and mPaxillin or the name of the gene pxna- in the figures and when addressing in the text - is just more clear.
- 4- Not clear for me what was the in vitro migration assay (scratch -could not find in the methods - sorry).
- 5- Not all figures say how many independent experiments were performed, please indicate.
- 6- Also, I think it would be fundamental and very enriching to measure FRAP (turnover / assemble/ disassemble and velocity) in vivo in another environment besides the skin - the CHT for example.
- 7- In vitro DOCK180 is not being recruited by paxillin but cells migrate more? Can there be an alternative pathway being activated? Can you discuss this further

- 8- When the same cells are placed in vitro vs in vivo, in vivo they are migrating more slowly. Or do you think this is an assay problem / variability (there is a lot of dispersion or a question of N)?
How do authors think this is working- what is the model here? Is there a correlation between velocity and low turnover (I thought it was high turnover more migration ?)- more time in contact migrate more efficiently? But then this goes against the macrophage results? I guess my question is velocity is not correlating with phosphorylation or presumably with turnover - what is the model? Or the turnover is independent of phosphorylation- please clarify. This is why I really think is important to check turnover and assembly/disassembly in the phosphorylation mutants.

November 29, 2022

Dear *Journal of Cell Biology* Editors,

Please find attached our revised manuscript entitled “Lack of Paxillin phosphorylation promotes single cell migration *in vivo*” for consideration as an Article in *JCB*. In this manuscript, we show the dynamics and regulation of focal adhesions (structures that link the cell cytoskeleton to the outside matrix) in migrating single cells in real-time in a living animal. To our knowledge, this is the first time focal adhesion dynamics are tracked and analyzed during single cell migration *in vivo*, and using this unique system, we have uncovered new mechanisms of focal adhesion regulation. We were delighted to receive positive reviews from the reviewers and the editors after the initial submission, and we have completed a number of experiments that address the main concerns and questions that were raised upon review, including additional FRAP and lifetime analysis of focal adhesions, as well as biochemical experiments to better dissect downstream signaling pathways. A description of our approaches and our point-by-point responses are detailed in the pages of this cover letter document (please see below). We feel that the inclusion of these data have greatly strengthened our central finding that lack of phosphorylation of a key focal adhesion component, Paxillin, leads to increased cell migration *in vivo*, hyperactivation of the CRKII-DOCK180/RacGEF pathway, and increased focal adhesion disassembly rates. This process is fundamentally different than what is expected from information generated in cell culture systems, revealing that focal adhesions are differentially regulated *in vivo* versus *in vitro* cell culture, and our results suggest a new mechanism for how focal adhesions are regulated during single cell migration.

We feel that our findings will be of significant interest to the greater scientific community of cell biology, developmental biology, and cancer. Given our results in cancer cells and immune cells, as well as conservation of mechanisms in zebrafish and mouse models, we think that our findings will be broadly applicable to many types of cell migration in development and disease, which will be of great interest to the community readership of *Journal of Cell Biology*.

Sincerely,

Minna Roh-Johnson, PhD
Assistant Professor
Department of Biochemistry, University of Utah
roh-johnson@biochem.utah.edu

Response to Reviewers:

Thank you to the reviewers and the editor for their thorough reviews. Below we respond to the main concerns summarized by the editor (the editor's comments are copied below), as well as provide a point-by-point response to each of the reviewers. We feel that the addition of these data strengthened the main points of the study, and we are grateful for the suggestions.

Comments from Editor:

As you will see, the reviewers are overall enthusiastic about this in vivo study, and we, the editors, feel the same. However, the reviewers raise some overlapping concerns that would need to be further addressed with new experimentation, which we agree with.

Reviewers #2 and #3 feel the functional relationship between the migration velocities, the dynamics of focal adhesions and the status of Paxillin(Y118) phosphorylation needs additional investigation, and thus they request that you examine the turnover/assembly/disassembly of focal adhesions in the phosphorylation mutants in vivo by FRAP assays. We acknowledge that quantification of dynamics is challenging in vivo, but these analyses would raise the impact of the paper. It is also particularly important, as noted by reviewer #2, that you address the level of expression of the constructs in vivo and report if you see a difference with the bright and dim cells.

In addition, reviewer #2 thinks more evidence is required to support the proposed mechanism in YUMM1.7 cells that lack of Paxillin(Y118) phosphorylation activates the CRKII-DOCK180-Rac pathway, which we agree with -for instance, the requirement of Paxillin(Y118) phosphorylation for CRKII binding and the activation of Rac signaling needs better demonstration.

Reviewers #1 and #3 also ask whether the described phenotypes are tissue-specific (would like that you study focal adhesions in a second tissue), and about the nature of the substrate that the cells are migrating in the living animal. Our view is that including a second tissue is not strictly required for resubmission, but the study of focal adhesions in zebrafish macrophages, for example, which you showed that exhibit enhanced migration in the absence of Paxillin(Y118) phosphorylation, is highly recommended for a successful revision.

Response to Editor's comments

We are delighted to hear that the editors and reviewers were "overall enthusiastic about this in vivo study". Below we respond to the main concerns raised in the editor's comments.

Main Concerns:

1. The functional relationship between the migration velocities, the dynamics of focal adhesions, and the status of Paxillin(Y118) phosphorylation need additional investigation (Reviewers #2 and #3). The reviewers request that we examine the turnover/assembly/disassembly of focal adhesions in the phosphorylation mutants in vivo by FRAP assays. We were also asked to address the levels of expression constructs in vivo.
2. The proposed mechanism in YUMM1.7 cells that lack of Paxillin(Y118) phosphorylation activates the CRKII-DOCK180-Rac pathway needs better demonstration (Reviewers #2 and #3). For instance, the requirement of Paxillin(Y118) phosphorylation for CRKII binding and the activation of Rac signaling needs better demonstration.

Of less concern:

3. It is unclear whether the described phenotypes are tissue-specific (Reviewers #1 and #3). The reviewers would like us to study focal adhesions in a second tissue and describe the nature of the substrate that the cells are migrating in the living animals. However, the editor noted that including a second tissue is not strictly required for resubmission, but would be recommended for a successful revision.

Response to Main Concern 1: Using ZMEL cells (zebrafish melanoma cells) expressing wildtype-Paxillin, the phosphomimetic Y118E-Paxillin, and the non-phosphorylatable Y118F-Paxillin, we transplanted these ZMEL cells into zebrafish larvae and first quantified mobility kinetics with FRAP assays. Paxillin mobility kinetics were

overall faster *in vivo* than in cell culture, consistent with our initial results. We found no significant differences in mobility kinetics between WT, Y118E, and Y118F-Paxillin *in vivo*. This result was expected as previous work has shown that mutations affecting the phosphorylation state of other focal adhesion proteins did not affect mobility kinetics (Stutchbury et al., 2017). Interestingly, in cell culture, we observed differences in mobility kinetics in cells in culture between WT, Y118E, and Y118F-Paxillin, with Y118E exhibiting faster $t_{1/2}$ values and Y118F exhibiting slower $t_{1/2}$ values, again revealing differences in regulation between *in vitro* and *in vivo* conditions. These data are incorporated into Figure 3F and Supplementary S4D.

We also sought to quantify overall focal adhesion lifetimes in cells expressing wildtype, Y118E, and Y118F-Paxillin *in vivo* and *in vitro*. The expression level of the constructs in these cells precluded the visualization of very small focal adhesion structures as the signal to noise was not robust enough. Thus, we could not accurately quantify the initiation of focal adhesion formation *in vivo*, therefore not allowing us to accurately quantify focal adhesion assembly. However, we were able to quantify the disassembly rates, as we were able to quantify the maximum fluorescence intensity for each structure, and then quantify the loss of fluorescence over time. When we analyzed disassembly rates in ZMEL cells expressing WT, Y118E, and Y118F-Paxillin in cell culture, we found that ZMEL cells expressing Y118E-Paxillin exhibited faster focal adhesion disassembly rates than Y118F-Paxillin expressing ZMEL cells. These results are consistent with the increased migration rate also observed in Y118E-Paxillin expressing ZMEL cells in culture compared to Y118F-Paxillin expressing ZMEL cells. Excitingly, when we analyzed disassembly rates *in vivo*, we found that ZMEL cells expressing the non-phosphorylatable Y118F-Paxillin also exhibited significantly higher focal adhesion disassembly rates than the phosphomimetic or wildtype version of Paxillin. These results are consistent with the observed increases in cell migration velocities when expressing the non-phosphorylatable Y118F-Paxillin *in vivo*, as increased focal adhesion turnover has been shown to lead to increased cell migration (Meenderink et al., 2010, Nagano et al., 2012, Webb et al., 2004). These data incorporated into Figure 3G.

Regarding the level of the expression constructs, all of the ZMEL Paxillin lines used in the study were FACS-isolated for equal Paxillin overexpression levels. We have included a western blot image to indicate the Paxillin expression levels of ZMEL cells compared to the parental ZMEL cells in Supplementary Figure S3A, as well as indicated this information in the text and the methods. The YUMM1.7 Paxillin cells were similarly FACS-isolated for equal Paxillin overexpression levels, as shown in Figure 5G,I,K,L (input lanes, showing Paxillin and/or GFP).

Response to Main Concern 2: Our initial submission provides evidence that expressing the non-phosphorylatable Y118F version of Paxillin led to increased cell migration *in vivo*. Furthermore, we tested the *requirement* for Y118-Paxillin phosphorylation in downstream interaction with CRKII and DOCK180/RacGEF by mutating this residue and assaying for Paxillin interactions with CRKII and DOCK180 with western blot analysis. We observed significantly increased interactions between Y118F-Paxillin and CRKII and DOCK180/RacGEF (Figure 5G-K), while observing modest interactions in cells in culture. From these results, we hypothesized that expression of Y118F-Paxillin leads to increased cell migration *in vivo* through hyperactivation of the CRKII/DOCK180/Rac pathway. However, the reviewer is correct in that we did not directly analyze Rac activity. We evaluated DOCK180/RacGEF because we specifically evaluated activation of pathways *at focal adhesions*, not global changes that may be occurring in the cell in response to expression of the Paxillin mutants. However, in response to the reviewers' comments, we did attempt to quantify Rac activity in cells expressing wildtype, Y118E, and Y118F-Paxillin *in vivo*. Using a Rac1-GTPase activity kit (Sigma Aldrich, 17-283), we immunoprecipitated PAK-1 PBD, which binds specifically to active GTP bound-Rac. In the cell culture conditions, we detected active Rac interacting with Paxillin (WT, Y118E, and Y118F) (Figure A below, top row). Furthermore, we detected total Rac in the crude lysate, with a slight increase in total Rac in cells expressing Y118E-Paxillin (Figure A, middle row), with GFP as a loading control (Figure A, bottom row). However, in the *in vivo* tumor samples, we could not detect active GTP-bound Rac in any of the conditions (Figure B, top lane), even though the *in vivo* positive control (adding GTP γ S) led to detectable GTP-bound Rac, indicating that the assay was performed correctly (Figure B, top row, last well). We reasoned that perhaps there were low levels of Rac in cells *in vivo*, and we therefore blotted whole cell lysate for total Rac and found that indeed, endogenous Rac levels are low in these cells (Figure B, 3rd row, upper band). When we loaded 7-times more protein input, we could begin to detect total endogenous Rac (Figure 4, 4th row). Thus, we then redid the experiment immunoprecipitating PAK-1 PBD, and probing for GTP-bound Rac, this time with 7-times more protein (7000 μ g) than our initial experiments. We still could not detect GTP-bound Rac under these

conditions (Figure B, 2nd row), despite observing a robust signal in our positive control (Figure B, 2nd row, last well). Thus, despite our best efforts, we were unable to directly evaluate Rac activity.

We then turned our attention to evaluating other relevant partners that could be activated downstream of Paxillin and CRKII, as this was also a suggestion made by Reviewer 2. Again, we specifically sought to test effectors that have been shown to be recruited to focal adhesions, and in addition to DOCK180/RacGEF, we also evaluated C3G/RasGEF. C3G/RasGEF has been shown to interact with CRKII and regulate cell adhesion and migration (Gotoh et al., 1995, Knudsen et al., 1994, Uemura and Griffin, 1999). Given that both of these GEFs also function at the leading edge of migrating cells, we sought to specifically test C3G interactions with Paxillin at focal adhesions. Using mouse melanoma tumors expressing either wildtype, phosphomimetic (Y118E), or non-phosphorylatable (Y118F) versions of mammalian Paxillin, we immunoprecipitated Paxillin, and found there is no difference in the level of C3G interactions with Paxillin when cells are expressing wildtype, Y118E, or Y118F-Paxillin *in vivo* (Supplementary Fig. S5D-I). We also assayed ERK activity, which is downstream of C3G activity, and found no difference in the phosphorylated active state of ERK across all conditions *in vivo* (Supplementary Fig. S5D-I). In contrast, we observed a dramatic enrichment of DOCK180/RacGEF interacting with non-phosphorylatable Y118F-Paxillin *in vivo* compared to cells expressing the phosphomimetic or wildtype Paxillin (Fig. 5K). Together, these results suggest that non-phosphorylatable Y118F-Paxillin exhibits increased cell migration *in vivo* through increased activation of the CRKII-DOCK180/RacGEF signaling pathway. Given that we could not directly assay Rac GTPase activity, we adjusted our language to not infer a role for Rac GTPase activity in this process, and instead focused on highlighting the activation of a RacGEF, but included the possibility of other pathway involvement in the discussion.

Response to Main Concern #3: We are grateful to the editor that inclusion of a secondary tissue was not strictly required for resubmission, as we were unable to visualize focal adhesion structures in migrating macrophages *in vivo*. We screened through a number of zebrafish, but found that if the GFP signal was too low, we could not visualize subcellular structures; and if too high, this resulted in high cytoplasmic signal that drowned out focal adhesion visualization, an issue that is common in zebrafish.

Given that we were unable to directly visualize focal adhesion structures in macrophages, we instead sought to address this concern in an indirect way. To address the question about whether described phenotypes are tissue-specific, we instead tested the functionality of Y118-Paxillin phosphorylation status in *another cell type* within the *same* tissue. If the Y118-Paxillin phosphostatus and its role in migration were largely driven by tissue cues, then we would expect the same migration phenotypes in Y118-Paxillin mutants in the same tissue, regardless of cell type. We specifically looked to analyze neutrophil migration as neutrophils migrate in the same tissue environments as macrophages.

We generated stable zebrafish lines in which wildtype, Y118F or Y118E-Paxillin was expressed in neutrophils using a neutrophil-specific promoter, *lyz*. We then evaluated overall neutrophil migration speeds using the same approaches as we used for macrophages by wounding larval tails and imaging directed neutrophil migration. We calculated neutrophil migration speeds by quantifying cell centroid displacement over time, and we found that expression of the non-phosphorylatable Y118F-Paxillin version did not increase neutrophil migration *in vivo*, as was observed in macrophages. Rather, expression of the non-phosphorylatable Y118F-

Paxillin led to reduced neutrophil migration *in vivo* compared to the phosphomimetic, and expression of the phosphomimetic Y118E-Paxillin led to increased neutrophil migration compared to the non-phosphorylatable Y118F-Paxillin (see figure below). These results suggest that first, the tissue environment does not solely dictate how Y118-Paxillin phosphorylation status affects cell migration, and that there are cell autonomous factors. Second, these results are exciting to us since neutrophils can adopt adhesion-independent migration in this tissue (Barros-Becker et al., 2017), and therefore, Paxillin phosphorylation leads to *opposite* phenotypes in cells migrating through adhesion-dependent versus adhesion-independent mechanisms.

We wanted to share these results with the reviewers, as we felt that the results partially addressed the question regarding tissue environments, albeit indirectly. We are currently incorporating the neutrophil data in a manuscript that examines how cells transition between mesenchymal (adhesion-dependent) and amoeboid (adhesion-independent) forms of motility *in vivo*. Thus, we did not include these data into this current manuscript, but would be willing to do so if the reviewers required it.

Point-by-point response to Reviewers:

Reviewer #1:

Bulk of studies addressing focal adhesion dynamics were performed using culture models where cells are plated in 2D and in some cases 3D culture models. Several years, a controversy of the existence of focal adhesions in 3D was resolved but there really had been no clear-cut answer to the molecular mechanisms that govern focal adhesions in vivo. This is a beautiful study that finally addresses the mechanism of focal adhesion formation for endogenously tagged proteins in a living animal. Using a suite of transgenic zebrafish lines, high quality intravital imaging and clever molecular tools, the authors determine that there is a differential regulation of tyrosine 118 in the assembly of focal adhesions. In culture models, tyrosine 118 is phosphorylated during adhesion formation in migrating cells. Here, the authors determined that this site shows a reduction in phosphorylation activity in functional adhesions that are formed during in vivo migration. The study is well done and the data are convincing.

I have really minor comments detailed below:

1) *For the in vivo imaging, it would be interesting to understand the chemistry/composition of the substrate that these cells are migrating. Staining for ECM markers can address this.*

-We thank this reviewer for their positive comments! Regarding the question about the substrates, we have included additional experiments evaluating extracellular matrix components. The original manuscript included laminin and collagen, but we have also included another extracellular matrix – fibronectin. Fibronectin variants have been shown to be expressed in skin, and there was an available antibody that has been shown to detect fibronectin in zebrafish. Zebrafish injected with ZMEL cells showed positive staining of fibronectin surrounding

those injected tumor cells (Supplementary Figure S1A). Interestingly, larvae that were not injected with ZMEL cells did not show positive fibronectin staining (Supplementary Figure S1A). These results that ZMEL cells might secrete fibronectin, or induce neighboring cells to secrete fibronectin, as has been shown in previous work (Barney et al., 2020, Lin et al., 2019, Rick et al., 2019). These data were incorporated into Supplementary Fig. S1A.

2) *It is intriguing that there is a differential phosphorylation of tyrosine 118, if cells are placed in a different organ environment where presumably the physi-chemical composition is different, would that finding be conserved. E.g. if you place in the tail/fin of the zebrafish would it show the same regulation. I understand that the macrophages show increased motility due to the reduced phosphorylation to the wounded area. Instead, I am asking, once the cells get to that environment is there a switch to a different focal adhesion mechanism or is it conserved for the in vivo environments tested.*

-Please see response to Main Comment 3 above.

Minor quibble: Graphs of motility can be shown as <msd>, where the plots can then be used to assess types of motility, persistence etc.

- Thank you for the suggestion. We have re-graphed migration as <msd>, and we have included these data and directionality data in Supplemental Figure S4A-C,E.

Reviewer #2:

In this manuscript, Qian Xue and colleagues explore the regulation and function of the focal adhesion paxillin during in vitro and in vivo migration. The past literature has pointed to important differences in focal adhesions' functions between in vivo 3D, soft environment migration and in vitro 2D, rigid substrate migration. This is a nice tentative to elucidate the molecular differences that are responsible for the generally weak adhesions observed in vivo. The authors show that paxillin Tyr118 phosphorylation is strongly reduced in the in vivo situation, possibly because of a low level of FAK. Moreover, they show that in these in vivo conditions, the low phosphorylation levels facilitates migration, whereas it inhibits 2D in vitro migration. They then attempt to determine how non-phosphorylatable paxillin could contribute to migration and propose that it activates the CRKII-DOCK180-Rac pathway. There are some interesting observations, however I found the last part of the manuscript less convincing first because it challenges the common idea that paxillin binding to CRKII requires Tyr118 phosphorylation to interact with the CRKII SH2 domain and therefore must be documented by some very strong convincing evidence. This is unfortunately not the case in particular because of technical imprecisions, which need to be clarified.

General comments

1- *What is the link between the difference in focal adhesion and paxillin dynamics observed in figure 2 and the phosphorylation status of paxillin? How do the phosphorylation mutant affect the dynamics of focal adhesions? Is the slower dynamics responsible for the slower migration? Although this hypothesis is tempting, the change in focal adhesion dynamics may also be a consequence of the slower migration observed.*

-Please see response to Main Concern 1 above.

2- *The proposed mechanism exposed remains very hypothetical at this stage and I am not sure figure 6 should be shown as many points are not actually demonstrated. The authors propose that FAK low expression level is responsible for the low level of phosphorylated paxillin. However, this is not demonstrated. Can FAK overexpression rescue paxillin phosphorylation?*

-We agree that the initial manuscript only showed low FAK expression levels and low levels of Paxillin phosphorylation. To determine whether these relationships are causal, we performed experiments suggested by this reviewer: We overexpressed FAK in YUMM1.7 cells and evaluated Paxillin phosphorylation levels in vivo. We found that indeed, overexpressing FAK rescued pY118-Paxillin phosphorylation in cells in vivo. These results suggest that FAK is capable of phosphorylating Y118-Paxillin in vivo, but that the low levels of phosphorylated Y118-Paxillin in cell in vivo is due to low endogenous levels of FAK. These data were incorporated in Figure 5E,F and Supplementary Figure S5B.

- *They also indicate that the non phosphorylated paxillin better activates the Rac GEF DOCK180. Can they show that Rac is overactivated?*

-Please see response to Main Concern 2. Due to our inability to directly measure Rac activity, we removed Rac from the pathway in Figure 6. However, we did bolster our findings by showing lack of change in parallel pathways.

- *The drawing of the cell indicate a similar number of focal adhesion in both in vivo and in vitro situation. This does not seem to be the case from the images shown, but as not been quantified.*

-Thank you for this suggestion. This reviewer is correct - We did not quantify the number of focal adhesions. We tried to do so, but did not feel that we could accurately count the number of total focal adhesions in cells in vivo due to the low signal to noise in parts of the cell (particularly the “back” of a cell migrating away from a mass). Thus, we took the drawing of focal adhesions out of Figure 6.

- *Do the authors have any indication of an increased recruitment of DOCK180 and/or CRKII at focal adhesions? Since the focal adhesions appear very small and limited in number in vivo compared to in vitro, an increase in Rac activity would required a very strong concentration of these proteins. Could this signaling take place somewhere else?*

-We apologize for the confusion, but yes, we do provide some evidence that suggests that the increased recruitment of DOCK180 and/or CRKII is at focal adhesion sites. In all of the CRKII and DOCK180/RacGEF western blot experiments, we first immunoprecipitated Paxillin and then blotted for the levels of CRKII or DOCK180/RacGEF that co-immunoprecipitated with Paxillin. We agree that a very strong concentration of these proteins at focal adhesions would likely be required to regulate cell migration; thus, we were quite struck by the dramatic increase in CRKII and DOCK180/RacGEF that co-immunoprecipitated with the non-phosphorylatable Y118F-Paxillin in vivo, suggesting hyperactivation of the pathways. We have revised the manuscript to make it clear that the experiments were not performed in bulk cell/tumor lysates.

3- *Paxillin is known to interact with multiple signaling molecules through both its phosphorylated tyrosines Y31 and Y118. Even if I understand that the authors do not want to study the functional role of Y31 in this paper, can they simply show if Y31 phosphorylation is also downregulated in vivo? Alternative, relevant partners of paxillin and CRKII such as p120Ras GAP or C3G are not considered or discussed. Since Rac overactivation is not shown, it is difficult to rule these out.*

-Thank you for the suggestion. We analyzed the phosphorylation status of Y31-Paxillin in vivo using YUMM1.7 tumors and found that similar to pY118-Paxillin, pY31-Paxillin was downregulated in vivo compared to in vitro. These data are included in Supplemental Figure 5J,K.

We have also evaluated other relevant partners and showed that their activity did not appear to change – Please see Response to Main Concern 2.

4- *One major concern is the relative expression level of GFP-paxillin constructs compared to the endogenous proteins in the different models.*

- *Since the constructs and the transfection protocols are different in the different settings, it would be important to show the complete Paxillin western blot (showing both the endogenous and the GFP tagged proteins) in figures. Moreover, for the mutant forms to have a dominant effect, there need to be a high level of expression compared to the endogenous.*

-Yes, the reviewer is correct in that for the mutant Paxillin forms to have a dominant effect, there needs to be a high expression level compared to endogenous Paxillin. We performed western blot analysis to show that the endogenous Paxillin protein level is low compared to the overexpression Paxillin protein level. The exogenous proteins are at least 10x higher in expression level than the endogenous Paxillin in the wildtype-Paxillin transfected cells. We also show that the expression level of each of the mutant constructs to show equal levels of expression across conditions (Supplemental Figure S3A).

- *Also, can the authors also be clarify where the GFP tag is? From the mat and met section, it seems that in some constructs it is located in C-ter (as expected) but may be in N-term in others. A N-term GFP tag is likely to affect the interactions with paxillin partners, such as CRKII.*

-Thank you for this suggestion - We have revised the manuscript to make the location of the GFP tag clear. For the mammalian experiments, since we are performing downstream biochemical analysis, we used YUMM1.7 cells that express Paxillin constructs with a cleavable T2A-GFP tag that is located in the C-terminus of Paxillin. We specifically chose the T2A-GFP tag not only because of the ease of sorting transduced cells with equal expression level, but also to avoid using a GFP fusion protein that may affect interactions of Paxillin to its partners.

Specific comments

1- *Figure 2. When characterizing the dynamics of focal adhesions, the authors should first quantify the number and size of focal adhesions, which is an important parameter to take into account when looking at the effect of FAK and paxillin activity.*

We have quantified the size of Paxillin punctae both in vivo and in the cell culture model, and the data are incorporated into Supplemental Figure S1D. Our quantifications show that Paxillin punctae are significantly smaller in vivo, and this is still likely to be an under-representation. Given that the size of focal adhesion in vivo is significantly smaller than the size in vitro and the low signal to noise made it such that we were likely to miss many smaller adhesions with our microscopy methods, we were unable to accurately quantify the total number of focal adhesions in vivo.

2- *Figure 3 and 4a. The immunofluorescence images in panel b are difficult to see. Zooms of specific regions of interest should be shown.*

-Thank you for the suggestion. We have included zoom panels.

3- *There is very little or no Phospho Y118 staining visible but this may be due to the very small and sparse focal adhesions observed in vivo. To further confirm the lack of phosphorylation, the authors should show P-Y118 staining together with total paxillin (GFP or immunofluorescence) or another marker of focal adhesions.*

-We found that similar to other in vivo models, the fixation process does not allow for ideal focal adhesion resolution, which is why we primarily used live imaging techniques to visualize focal adhesion structures in vivo. However, we were able to evaluate pY118-Paxillin and total Paxillin levels with western blot analysis in the in vivo mouse models (Figure 3C-D). The ratiometric analysis of pY118-Paxillin/Paxillin suggest that Paxillin phosphorylation levels are significantly lower in vivo despite comparable levels of total Paxillin in vivo versus in culture.

4- *Line 268. The sentence suggests that the non-phosphorylatable Paxillin reduces cell velocity compared to the wild type, whereas the graph shows that the velocity is decreased only when compared to the phospho-mimetic mutant. The authors cannot conclude (line 274) that the "non-phosphorylatable Y118 paxillin inhibits cell migration"*

-Thank you for the suggestion. We have revised this sentence.

5- *Figure 5. I am confused as to why is the total paxillin immunoprecipitated and not only the mutants. It is important to show the endogenous and GFP tagged paxillin in all the figure panels. Or are the mutants untagged here? In this case, the authors should redo the experiment with tagged version and GFP IP.*

We apologize for the confusion – We did not make this aspect clear in the text, which we have remedied in the resubmission. The exogenous Paxillin proteins are at least 10x higher in expression level than the endogenous Paxillin in the wildtype-Paxillin transfected cells (Figure 3 C, Paxillin expression in parental lane compared to experiment lanes) and will therefore have a dominant negative effect. Since we are performing downstream biochemical analysis, we used YUMM1.7 cells that express Paxillin constructs with a cleavable T2A-GFP tag that is located in the C-terminus of Paxillin. We specifically chose the T2A-GFP tag not only because of the ease of sorting transduced cells with equal expression level, but also to avoid using a GFP fusion protein that may affect interactions of Paxillin to its partners. Thus, we immunoprecipitated Paxillin, not GFP, and the Paxillin mutant proteins exhibit a 10X higher level of expression than endogenous Paxillin.

6- *Fig 5. Panel d. What is FAK level normalized to? If it is to GFP, this does not seem the most appropriate. Another blot showing actin for instance could be used.*

We went back and forth quite a bit for this quantification to determine how to normalize the FAK levels and consulted with our biochemistry colleagues. After discussions, we opted for GFP for the following reasons: 1) We were cautious to use actin for normalization due to the changes in cell migration rates in vivo versus in culture; 2) We wanted to control for the number of cancer cells in our in vivo condition given that we used tumors as lysates (ie. tumors contain a large number of stromal cells in addition to the GFP-positive tumor cells). The GFP levels allow us to compare similar numbers of cancer cells in vivo versus in cell culture, which we could not do with actin levels. For these reasons, the normalization for FAK levels was performed with GFP, not actin.

7- Figure 5. Quantification based on only 2 technical repeats is insufficient and cannot be used for statistical test. Biological repeats and more repeats in general are required.

-We apologize for the confusion. All experiments in Figure 5 were performed at least 3 times. When "2 technical replicates" was listed, it meant that the experiment was performed at least 3 times, each time with 2 technical replicates (although, most times we used more than 2 technical replicates), and the biological replicates are quantified in the graphs. With the additional experiments performed for the resubmission, we have increased our "n" across the board, and have made this information clearer in the figure legend.

8- For all graphs, please indicate the exact p values instead of the stars.

-We have made this change.

9- For the western blot quantifications, the data corresponding to each experimental values should be shown in the graph (not only the mean+ error bars : s.d. or SEM?)

- We have made this change.

10- I think that for clarity, the different pieces of a same western blot (for instance in 5b) should be shown as a single panel with a single line indicating the cut region. As it is, it appears as if coming from distinct blots and therefore not quantifiable.

-Thank you for the suggestion. For all blots that were cut, we have included all uncut/unmodified blots as supplemental data, and we made this information clear by specifically indicating "unmodified blots are in Supplemental Figure X" in the figure legends.

Reviewer #3:

1. A short summary of the paper, including description of the advance offered to the field. If you feel that prior literature undermines any aspect of this work, specific references are appreciated.

Really interesting study that highlights fundamental molecular differences between in vivo and in vitro mechanisms of cell migration, challenging the dogma that has been created by in vitro studies, showing clearly that in vivo studies are fundamental.

The authors developed a zebrafish transplantation system to visualize focal adhesion structures during single cell migration in vivo with high-resolution live cell imaging and compared focal adhesion dynamics to the traditional in vitro cell culture model.

By FRAP and monitoring the fluorescence recovery over time the authors found that Paxillin exhibits a significantly faster molecular turnover rate in vivo as compared to cells in vitro. Also, the assembly and disassembly rates differed between the in vivo and in vitro environments (in vivo higher assembly and reduced disassembly).

Then the authors "sought to identify the molecular regulation that might explain these differences" and found that Paxillin, tyrosine 118 (Y118), exhibits reduced phosphorylation in migrating cells in vivo in both zebrafish and mouse melanoma models, contrary to the pivotal role for this phosphorylation event in cell culture studies. Modulation of this residue by site-directed mutagenesis leads to opposite cell migration phenotypes in vivo versus in vitro in both migrating cancer cells and macrophages (reduced phosphorylation in vivo increases cell velocity but in vitro it reduces velocity of the cells). FAK is downregulated in cells in vivo, but the downstream effectors DOK180 and CRKII are upregulated. Cells expressing non-phosphorylatable Y118-Paxillin exhibit increased activation in CRKII-DOCK180-Rac pathway, leading to increased cell migration.

2. For each main point of the paper, please indicate if the data are strongly supportive. If not, indicate the evidence you feel is required. Please highlight those points you feel are most crucial for further consideration at JCB.

By FRAP and monitoring the fluorescence recovery over time the authors found that Paxillin exhibits a significantly faster molecular turnover rate in vivo as compared to cells in vitro. Also, the assembly and

disassembly rates differed between the *in vivo* and *in vitro* environments (*in vivo* higher assembly and reduced disassembly). OK

Then the authors "sought to identify the molecular regulation that might explain these differences" and found that Paxillin, tyrosine 118 (Y118), exhibits reduced phosphorylation in migrating cells *in vivo* in both zebrafish and mouse melanoma models, contrary to the pivotal role for this phosphorylation event in cell culture studies. (Fig.3). Modulation of this residue by site-directed mutagenesis leads to opposite cell migration phenotypes *in vivo* versus *in vitro* in both migrating cancer cells and macrophages (reduced phosphorylation *in vivo* increases cell velocity but *in vitro* it reduces velocity of the cells). OK

I think this is where I found data was missing (or maybe I missed) showing that the non-phosphorylatable residue Y118F indeed had a >> turnover/assembly rate (could see this *in vitro*), or the constitutively active tyrosine phosphorylation (Y118E) had lower turnover (*in vivo*?).

-Please see our response to Main Concern 1.

FAK is downregulated in cells *in vivo*, but the downstream effectors DOK180 and CRKII are upregulated. Cells expressing non-phosphorylatable Y118-Paxillin exhibit increased activation in CRKII-DOCK180-Rac pathway, leading to increased cell migration. OK

3.Lastly, indicate any additional issues you feel should be addressed (text changes, data presentation, etc.)

Comments: As a non-expert in cell migration, I apologize if some points were not clear to me.

1- By FRAP and monitoring the fluorescence recovery over time the authors found that Paxillin exhibits a significantly faster molecular turnover rate *in vivo* as compared to cells *in vitro*. Also, the assembly and disassembly rates differed between the *in vivo* and *in vitro* environments (*in vivo* higher assembly and reduced disassembly).

For me was not so clear how you have a faster molecular turnover rate but have lower disassembly rate - can you please clarify?

-The molecular turnover rate of Paxillin is mechanistically distinct from assembly/disassembly rates of Paxillin-positive structures. The molecular turnover rate as measured by FRAP analysis measures the diffusion properties of Paxillin at focal adhesion sites. The recovery of Paxillin fluorescence at the bleached site is due to the exchange of bleached molecules with unbleached Paxillin molecules from the surrounding area, so this measurement provides information about how Paxillin interacts with the focal adhesion complex. Separately, assembly and disassembly rates of Paxillin-positive structures quantifies the change in Paxillin fluorescence over time at a specific focal adhesion site over the lifetime of the Paxillin-positive structure. This quantification will tell you how fast the Paxillin-positive structure grows to its maximum size and shrinks to the point of non-detection. While the molecular turnover of a protein can have an effect on overall focal adhesions lifetimes, that is not always the case. A protein can exhibit fast diffusion rates in and out of the structure (faster molecular turnover), but the structure itself can disassemble slowly, because these two metrics can be uncoupled. We have included a brief comment in the discussion to better reflect this point.

2- In figure 3b, it would be nice to show that that the Ab does not recognize the Y118F as a control - since what I understood this Ab was raised against mammalian Paxillin (and in this figure it is the zebrafish Paxillin- or *pxna*-that is being overexpressed).

-Sorry for the confusion. This figure is showing endogenous levels of Paxillin, and the GFP expression is a space fill GFP in ZMEL cells. There is no overexpression of any forms of Paxillin in Figure 3B.

However, we do think that it is important to show that the antibody does not recognize the mutant forms of Y118-Paxillin, and we have included a western blot in Supplemental Figure S2B to show this result.

3- It is not clear throughout the manuscript if the authors are talking about the zebrafish paxillin, mouse or human. I had to go through the methods/legends..it would be useful to put zPaxillin and mPaxillin or the name of the gene *pxna*- in the figures and when addressing in the text - is just more clear.

-We made clarified whether we are referring to zebrafish versus mammalian Paxillin throughout the text and figure legends. Thanks for the suggestion.

4- Not clear for me what was the *in vitro* migration assay (scratch -could not find in the methods - sorry).

-The *in vitro* migration assay was performed by quantifying centroid displacement of migrating cells in culture. We have included this description in the methods section.

5- Not all figures say how many independent experiments were performed, please indicate.
-We have now indicated the number of experiments in each figure legend.

6- Also, I think it would be fundamental and very enriching to measure FRAP (turnover / assemble/ disassemble and velocity) in vivo in another environment besides the skin - the CHT for example.
-Please see our response to Main Concern #1 and #3 above.

7- In vitro DOCK180 is not being recruited by paxillin but cells migrate more? Can there be an alternative pathway being activated? Can you discuss this further?

-There is DOCK180 being recruited to Paxillin in vitro. In Figure 5K, the “absence” of DOCK180 recruitment to Paxillin in vitro is because we had to treat the in vitro and in vivo samples the same (ie. they were on the same blot) to make comparisons about the DOCK180 recruitment to Paxillin between these conditions. Given the increased DOCK180 recruitment to Paxillin observed in vivo, the signal in vitro is very low. However, we cut and re-exposed just the in vitro samples to show that indeed, there is DOCK180 recruited to Paxillin in vitro, as would be expected (see example image below):

8- When the same cells are placed in vitro vs in vivo, in vivo they are migrating more slowly. Or do you think this is an assay problem / variability (there is a lot of dispersion or a question of N)?
How do authors think this is working- what is the model here? Is there a correlation between velocity and low turnover (I thought it was high turnover more migration ?)- more time in contact migrate more efficiently? But then this goes against the macrophage results? I guess my question is velocity is not correlating with phosphorylation or presumably with turnover - what is the model? Or the turnover is independent of phosphorylation- please clarify. This is why I really think is important to check turnover and assembly/disassembly in the phosphorylation mutants.

-I think there is some confusion regarding Paxillin turnover/diffusion as quantified by FRAP assays versus assembly/disassembly kinetics of Paxillin-positive structures. We hope we have clarified this point above. However, you raise an important question regarding what aspect of focal adhesion dynamics explains the change in migration, and how do the Paxillin mutants affect these focal adhesion dynamics. To address this concern, please see our response to Main Concern #1 above.

References:

- BARNEY, L. E., HALL, C. L., SCHWARTZ, A. D., PARKS, A. N., SPARAGES, C., GALARZA, S., PLATT, M. O., MERCURIO, A. M. & PEYTON, S. R. 2020. Tumor cell-organized fibronectin maintenance of a dormant breast cancer population. *Sci Adv*, 6, eaaz4157.
- BARROS-BECKER, F., LAM, P. Y., FISHER, R. & HUTTENLOCHER, A. 2017. Live imaging reveals distinct modes of neutrophil and macrophage migration within interstitial tissues. *J Cell Sci*, 130, 3801-3808.
- GOTOH, T., HATTORI, S., NAKAMURA, S., KITAYAMA, H., NODA, M., TAKAI, Y., KAIBUCHI, K., MATSUI, H., HATASE, O., TAKAHASHI, H. & ET AL. 1995. Identification of Rap1 as a target for the Crk SH3 domain-binding guanine nucleotide-releasing factor C3G. *Mol Cell Biol*, 15, 6746-53.
- KNUDSEN, B. S., FELLER, S. M. & HANAFUSA, H. 1994. Four proline-rich sequences of the guanine-nucleotide exchange factor C3G bind with unique specificity to the first Src homology 3 domain of Crk. *J Biol Chem*, 269, 32781-7.
- LIN, T. C., YANG, C. H., CHENG, L. H., CHANG, W. T., LIN, Y. R. & CHENG, H. C. 2019. Fibronectin in Cancer: Friend or Foe. *Cells*, 9.
- MEENDERINK, L. M., RYZHOVA, L. M., DONATO, D. M., GOCHBERG, D. F., KAVERINA, I. & HANKS, S. K. 2010. P130Cas Src-binding and substrate domains have distinct roles in sustaining focal adhesion disassembly and promoting cell migration. *PLoS One*, 5, e13412.
- NAGANO, M., HOSHINO, D., KOSHIKAWA, N., AKIZAWA, T. & SEIKI, M. 2012. Turnover of focal adhesions and cancer cell migration. *Int J Cell Biol*, 2012, 310616.
- RICK, J. W., CHANDRA, A., DALLE ORE, C., NGUYEN, A. T., YAGNIK, G. & AGHI, M. K. 2019. Fibronectin in malignancy: Cancer-specific alterations, protumoral effects, and therapeutic implications. *Semin Oncol*, 46, 284-290.
- STUTCHBURY, B., ATHERTON, P., TSANG, R., WANG, D. Y. & BALLESTREM, C. 2017. Distinct focal adhesion protein modules control different aspects of mechanotransduction. *J Cell Sci*, 130, 1612-1624.
- UEMURA, N. & GRIFFIN, J. D. 1999. The adapter protein Crk1 links Cbl to C3G after integrin ligation and enhances cell migration. *J Biol Chem*, 274, 37525-32.
- WEBB, D. J., DONAIS, K., WHITMORE, L. A., THOMAS, S. M., TURNER, C. E., PARSONS, J. T. & HORWITZ, A. F. 2004. FAK-Src signalling through paxillin, ERK and MLCK regulates adhesion disassembly. *Nat Cell Biol*, 6, 154-61.

December 28, 2022

RE: JCB Manuscript #202206078R

Minna Roh-Johnson
University of Utah
Biochemistry
15N Medical Drive East
Salt Lake City, UT 84112-5650

Dear Dr. Roh-Johnson:

Thank you for submitting your revised manuscript entitled "Lack of Paxillin phosphorylation promotes single cell migration in vivo". The three original reviewers have now assessed your revised manuscript and, as you can see, they are satisfied with revisions. Thus, we would be happy to publish your paper in JCB pending final revisions necessary to meet our formatting guidelines (see details below).

To avoid unnecessary delays in the acceptance and publication of your paper, please read the following information carefully. Please go through all the formatting points paying special attention to those marked with asterisks.

A. MANUSCRIPT ORGANIZATION AND FORMATTING:

1) Text limits: Character count for Articles and Tools is < 40,000, not including spaces. Count includes title page, abstract, introduction, results, discussion, and acknowledgments. Count does not include materials and methods, figure legends, references, tables, or supplemental legends.

2) Figures limits: Articles and Tools may have up to 10 main text figures.

***** Please note that main text figures should be provided as individual, editable files.**

3) Figure formatting:

***** Molecular weight or nucleic acid size markers must be included on all gel electrophoresis. Please, include MW markers on gels in Figs 3C, 5B, 5E, 5G, 5I, 5K-L, S2B, S2D, S3A, S5A-D, S5G, and S5J.**

***** Scale bars must be present on all microscopy images, including inset magnifications. Please add scale bars to Fig 1H (inset magnifications).**

Also, please avoid pairing red and green for images and graphs to ensure legibility for color-blind readers. If red and green are paired for images, please ensure that the particular red and green hues used in micrographs or graphs are distinctive with any of the colorblind types. If not, please modify colors accordingly or provide separate images of the individual channels.

4) Statistical analysis:

Error bars on graphic representations of numerical data must be clearly described in the figure legend.

***** The number of independent data points (n) represented in a graph must be indicated in the legend -please add 'n' to legend of Fig 4D, 5M (revise 'n' for 5N, there are four dots in graphs but 'n' reads as "3 individual experiments", and indicate whether 'n' refers to technical or biological replicates (i.e. number of analyzed cells, samples or animals, number of independent experiments).**

If independent experiments with multiple biological replicates have been performed, we recommend using distribution-reproducibility SuperPlots (please, see Lord et al., JCB 2020) to better display the distribution of the entire dataset, and report statistics (such as means, error bars, and P values) that address the reproducibility of the findings.

Statistical methods should be explained in full in the materials and methods in a separate section.

For figures presenting pooled data the statistical measure should be defined in the figure legends.

Please also be sure to indicate the statistical tests used in each of your experiments (both in the figure legend itself and in a separate methods section) as well as the parameters of the test (for example, if you ran a t-test, please indicate if it was one- or two-sided, etc.).

If you used parametric tests in your study (i.e. t-tests), you should have first determined whether the data was normally distributed before selecting that test. In the stats section of the methods, please indicate how you tested for normality. If you did not test for normality, you must state something to the effect that "Data distribution was assumed to be normal but this was not formally tested."

5) Abstract and title:

The abstract should be no longer than 160 words and should communicate the significance of the paper for a general audience.

The title should be less than 100 characters including spaces. Make the title concise but accessible to a general readership.

6) Materials and methods:

Should be comprehensive and not simply reference a previous publication for details on how an experiment was performed. The text should not refer to methods "...as previously described."

Also, the materials and methods should be included with the main manuscript text and not in the supplementary materials.

7) For all cell lines, vectors, constructs/cDNAs, etc. - all genetic material: please include database / vendor ID (e.g., Addgene, ATCC, etc.) or if unavailable, please briefly describe their basic genetic features, even if described in other published work or gifted to you by other investigators (and provide references where appropriate).

Please be sure to provide the sequences for all of your oligos: primers, si/shRNA, RNAi, gRNAs, etc. in the materials and methods.

You must also indicate in the methods the source, species, and catalog numbers/vendor identifiers (where appropriate) for all of your antibodies, including secondary. If antibodies are not commercial, please add a reference citation if possible.

8) Microscope image acquisition:

The following information must be provided about the acquisition and processing of images:

- a. Make and model of microscope
- b. Type, magnification, and numerical aperture of the objective lenses
- c. Temperature
- d. imaging medium
- e. Fluorochromes
- f. Camera make and model
- g. Acquisition software
- h. Any software used for image processing subsequent to data acquisition. Please include details and types of operations involved (e.g., type of deconvolution, 3D reconstitutions, surface or volume rendering, gamma adjustments, etc.).

10) Supplemental materials:

There are strict limits on the allowable amount of supplemental data. Articles/Tools may have up to 5 supplemental figures. There is no limit for supplemental tables.

*** Please note that supplemental figures and tables should be provided as individual, editable files.

*** A summary of all supplemental material should appear at the end of the Materials and Methods section (please see any recent JCB paper for an example of this summary).

11) Video legends: Should describe what is being shown, the cell type or tissue being viewed (including relevant cell treatments, concentration and duration, or transfection), the imaging method (e.g., time-lapse epifluorescence microscopy), what each color represents, how often frames were collected, the frames/second display rate, and the number of any figure that has related video stills or images.

12) eTOC summary:

A ~40-50 word summary that describes the context and significance of the findings for a general readership should be included on the title page.

*** The statement should be written in the present tense and refer to the work in the third person. It should begin with "First author name(s) et al..." to match our preferred style.

13) Conflict of interest statement:

JCB requires inclusion of a statement in the acknowledgements regarding competing financial interests. If no competing financial interests exist, please include the following statement: "The authors declare no competing financial interests."

14) Author contribution section:

A separate author contribution section is required following the Acknowledgments in all research manuscripts.

*** All authors should be mentioned and designated by their first and middle initials and full surnames and the CRediT nomenclature is encouraged (<https://casrai.org/credit/>).

15) ORCID IDs: ORCID IDs are unique identifiers allowing researchers to create a record of their various scholarly contributions in a single place. At resubmission of your final files, please consider providing an ORCID ID for as many contributing authors as possible.

16) Materials and data sharing:

All animal and human studies must be conducted in compliance with relevant local guidelines, such as the US Department of Health and Human Services Guide for the Care and Use of Laboratory Animals or MRC guidelines, and must be approved by the authors' Institutional Review Board(s). A statement to this effect with the name of the approving IRB(s) must be included in the Materials and Methods section.

*** As a condition of publication, authors must make protocols and unique materials (including, but not limited to, cloned DNAs; antibodies; bacterial, animal, or plant cells; and viruses) described in our published articles freely available upon request by researchers, who may use them in their own laboratory only. All materials must be made available on request and without undue delay. We strongly encourage to deposit all the cell lines/strains and reagents generated in this study in public repositories.

All datasets included in the manuscript must be available from the date of online publication, and the source code for all custom computational methods, apart from commercial software programs, must be made available either in a publicly available database or as supplemental materials hosted on the journal website. Numerous resources exist for data storage and sharing (see Data Deposition: <https://rupress.org/jcb/pages/data-deposition>), and you should choose the most appropriate venue based on your data type and/or community standard. If no appropriate specific database exists, please deposit your data to an appropriate publicly available database.

17) Please note that JCB now requires authors to submit Source Data used to generate figures containing gels and Western blots with all revised manuscripts. This Source Data consists of fully uncropped and unprocessed images for each gel/blot displayed in the main and supplemental figures. The Source Data files will be directly linked to specific figures in the published article.

Since your paper includes cropped gel and/or blot images, please be sure to provide one Source Data file for each figure that contains gels and/or blots along with your revised manuscript files. File names for Source Data figures should be alphanumeric without any spaces or special characters (i.e., SourceDataF#, where F# refers to the associated main figure number or SourceDataFS# for those associated with Supplementary figures). The lanes of the gels/blots should be labeled as they are in the associated figure, the place where cropping was applied should be marked (with a box), and molecular weight/size standards should be labeled wherever possible.

Source Data Figures should be provided as individual PDF files (one file per figure). Authors should endeavor to retain a

minimum resolution of 300 dpi or pixels per inch. Please review our instructions for export from Photoshop, Illustrator, and PowerPoint here: <https://rupress.org/jcb/pages/submission-guidelines#revised>

B. FINAL FILES:

Thank you for this interesting contribution, we look forward to publishing your paper in Journal of Cell Biology.

Sincerely,

Anna Huttenlocher
Monitoring Editor
Journal of Cell Biology

Lucia Morgado-Palacin, PhD
Scientific Editor
Journal of Cell Biology

Reviewer #1 (Comments to the Authors (Required)):

Thank you for addressing my comments.

Reviewer #2 (Comments to the Authors (Required)):

I thank the authors for their answers to my comments. Although they have not been able to answer some of my concerns for technical reasons, they have added required and interesting pieces of data, clarified their conclusions and clearly improved the

manuscript. Overall, I think that the manuscript is now suitable for publication in the Journal of Cell Biology.

Reviewer #3 (Comments to the Authors (Required)):

The authors answered to most of my concerns.